# Revisiting Generative Infrared and Visible Image Fusion Based on Human Cognitive Laws

**Lin Guo[1]    Xiaoqing Luo[1*]    Wei Xie [1]    Zhancheng Zhang[2]**
**Hui Li[1]    Rui Wang[1]    Zhenhua Feng[1]    Xiaoning Song[1]**

[1]School of Artificial Intelligence and Computer Science, Jiangnan University, Wuxi, China
[2]School of Electronic and Information Engineering
Suzhou University of Science and Technology, Suzhou, China
`{guolin, xiewei}@stu.jiangnan.edu.cn`
`{xqluo, lihui.cv, cs_wr, fengzhenghua, x.song}@jiangnan.edu.cn`
`{zczhang}@usts.edu.cn`

## Abstract

Existing infrared and visible image fusion methods often face the dilemma of balancing modal information. Generative fusion methods reconstruct fused images by learning from data distributions, but their generative capabilities remain limited. Moreover, the lack of interpretability in modal information selection further affects the reliability and consistency of fusion results in complex scenarios. This manuscript revisits the essence of generative image fusion under the inspiration of human cognitive laws and proposes a novel infrared and visible image fusion method, termed HCLFuse. First, HCLFuse investigates the quantification theory of information mapping in unsupervised fusion networks, which leads to the design of a multi-scale mask-regulated variational bottleneck encoder. This encoder applies posterior probability modeling and information decomposition to extract accurate and concise low-level modal information, thereby supporting the generation of high-fidelity structural details. Furthermore, the probabilistic generative capability of the diffusion model is integrated with physical laws, forming a time-varying physical guidance mechanism that adaptively regulates the generation process at different stages, thereby enhancing the ability of the model to perceive the intrinsic structure of data and reducing dependence on data quality. Experimental results show that the proposed method achieves state-of-the-art fusion performance in qualitative and quantitative evaluations across multiple datasets and significantly improves semantic segmentation metrics. This fully demonstrates the advantages of this generative image fusion method, drawing inspiration from human cognition, in enhancing structural consistency and detail quality. The source code is available at https://github.com/lxq-jnu/HCLFuse

## 1   Introduction

Infrared and visible image fusion has been extensively employed as a core technique in multi-modal sensing systems, which are widely adopted in surveillance, autonomous driving, and target tracking [1, 2, 3]. Infrared sensors capture thermal radiation and are effective under low-light conditions, while visible sensors perform better in well-lit environments but degrade in darkness or adverse weather. Image fusion aims to combine complementary advantages from both modalities, traditionally formulated as a deterministic mapping based on handcrafted features, using techniques like multi-scale decomposition[4, 5, 6] or sparse representation[7, 8, 9]. Although these methods are computationally efficient, they lack the ability to capture semantic relationships and handle uncertainty, which often results in suboptimal integration of multi-modal information. Recent deep learning-based approaches adopt data-driven strategies to automatically learn cross-modal

39th Conference on Neural Information Processing Systems (NeurIPS 2025).

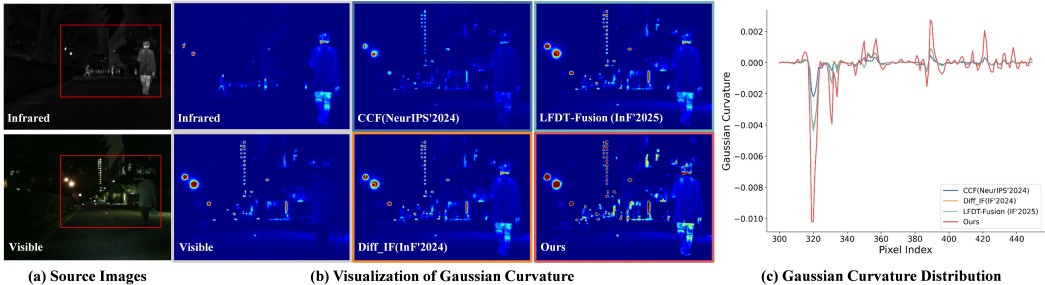

| (a) Source Images | (b) Visualization of Gaussian Curvature | (c) Gaussian Curvature Distribution |

Figure 1: Comparative visualization of gaussian curvature in generative infrared and visible image fusion methods.

relationships, substantially improving fusion quality. Generative models further advance this by modeling the task as a conditional distribution $p(z \mid x, y)$, capable of handling uncertainty and capturing deeper modality interactions. However, existing generative methods still face several limitations:

**Limited generative capability:** Existing methods often focus on feature extraction (representative methods: Dif-Fusion[10], LFDT-Fusion[11]) or optimization (representative methods: Diff-IF[12]), failing to fully leverage the potential of generative models.

**Lack of interpretability:** Current methods lack sufficient interpretability regarding how information from different modalities is selected and processed during fusion.

**Strong data dependence:** These models heavily rely on statistical distribution properties and lack a deep understanding of the intrinsic rules of the modalities. As a result, their robustness is weak when confronted with data distribution shifts or noise interference.

To visualize these limitations, Gaussian curvature is introduced as a geometric indicator of structural consistency. As shown in Fig. 1(b), existing diffusion-based methods exhibit fragmented curvature patterns in critical regions, suggesting incoherent structure retention and biased information integration. This indicates that the methods fail to achieve optimal information selection and retention in the integration of modal information. Existing generative models focus primarily on data distribution, neglecting the intrinsic understanding of the data. In contrast, human cognition inspires us to combine empirical data with abstract reasoning and domain knowledge. As highlighted by Tenenbaum et al.[13], mechanisms such as selective attention and physical laws play a crucial role in guiding robust perception under uncertainty, which remains largely absent in current fusion models. Selective attention involves prioritizing task-relevant information while disregarding redundant or irrelevant inputs. Adherence to physical laws involves integrating perceptual input with domain knowledge to support robust inference.

This paper revisits generative infrared and visible image fusion through cognition-inspired modeling principles and proposes a novel method. The framework(see Fig. 2(a)) integrates a data-driven generative model with theorem-constrained probabilistic reasoning, enabling more interpretable and robust fusion. A multi-scale variational bottleneck encoder is designed to extract structured low-level features through information quantization, which are then guided by a physics-aware diffusion process. By incorporating physical laws into the generative trajectory, HCLFuse enhances semantic consistency and reduces reliance on high-quality data. As shown in Fig. 1, HCLFuse not only demonstrates more complete structural features in curvature visualization but also surpasses the curvature quality of the original images in certain regions. Overall, the main contributions of this paper are summarized as follows:

- A novel generative fusion framework is proposed to enhance modality interpretability and structural consistency by incorporating human cognitive laws.
- A multi-scale variational bottleneck encoder is introduced to extract discriminative low-level representations through unsupervised information mapping quantization theory.
- The generative ability of the diffusion model is combined with physical laws to form a time-varying physical guidance mechanism, enhancing the ability of the model to perceive the intrinsic nature of data, reducing data dependence.

- Extensive experiments demonstrate superior performance in both fusion quality and downstream semantic segmentation.

## 2 Related Work

### 2.1 Deep Learning-Based Fusion Methods

With the advancement of deep learning, image fusion has evolved from traditional deterministic models to data-driven approaches capable of capturing complex modality relationships. Early methods such as DenseFuse[14] introduced convolutional encoders with dual fusion strategies, while NestFuse[15] enhanced detail preservation via multi-scale nested connections. To improve interpretability, LRRNet[16] employed a lightweight architecture that approximates optimal fusion solutions, and MMAE[17] incorporated masked attention mechanisms into a general two-stage fusion pipeline. Recent trends favor end-to-end architectures. PMGI[18] unified diverse fusion objectives under a common optimization formulation, and STDFusionNet[19] leveraged salient target masks to jointly model detection and fusion. Transformer-based methods further expanded global context modeling: SwinFusion[20] designed intra- and inter-domain modules based on the Swin Transformer; SegMiF[21] applied hierarchical attention for fine-grained representation; STFNet[22] focused on pixel-level dependencies to mitigate ghosting; and CrossFuse[23] proposed complementary attention to suppress modality-specific redundancy. While these methods have improved cross-modal feature integration, they often treat fusion as deterministic overlay, overlooking its generative nature. Consequently, their performance is limited when handling degraded inputs or incomplete modality information, restricting the semantic expressiveness of fused outputs.

### 2.2 Generative Models for Image Fusion

With the evolution of generative models, image fusion task has gradually been formulated within the framework of Generative Adversarial Networks (GAN). FusionGAN[24] was among the earliest attempts to establish an adversarial learning scheme between a generator and a discriminator, wherein the generator synthesized fused image and the discriminator evaluated the detail-level differences between the fused image and the visible image. This setup was intended to improve the structural integrity and perceptual realism of the fused result. The application of GAN in image fusion is hindered by inherent limitations, including training instability and mode collapse, which reduce both generalization ability and generation quality. To address these limitations, diffusion models(DM) have been introduced as a compelling alternative, offering advantages such as progressive generation, high fidelity, and stable training dynamics. Dif-Fusion[10] employed a diffusion model as a feature extractor to guide the fusion process and produced fused images with improved color fidelity. Diff-IF[12] incorporated prior knowledge of the fusion task to condition the diffusion model, enabling the generation of high-quality fused image even in the absence of ground-truth. CCF[25] proposed a conditionally controllable fusion framework that relaxed the constraints imposed by fixed fusion paradigms and improved the adaptability and generalization of the generation process.

Although diffusion-based image fusion methods have demonstrated notable advances in both performance and representational capability, most existing approaches continue to rely heavily on conditioning from data distribution. Therefore, the full potential of diffusion models as generative mechanisms for image fusion remains underexplored. This work establishes a theoretical framework for unsupervised image fusion from the perspective of human cognitive principles. Furthermore, a time-varying physical guidance mechanism is developed by integrating physical laws with data-driven distributions, aiming to generate fused images with improved structural consistency and information completeness.

## 3 Method

### 3.1 Problem Statement and Modeling

In HCLFuse, let the infrared image domain be denoted as $\mathcal{X}$ and the visible image domain as $\mathcal{Y}$, with joint observations $(x, y) \sim p_{x,y}$. A fusion mapping $\mathcal{F}(\mathcal{X}, \mathcal{Y}) \to \mathcal{Z}$ is constructed to generate a fused latent representation $z = \mathcal{F}(x, y) \in \mathcal{Z}$, where $\mathcal{Z} \subseteq \mathbb{R}^d$ denotes the fusion space. The detailed

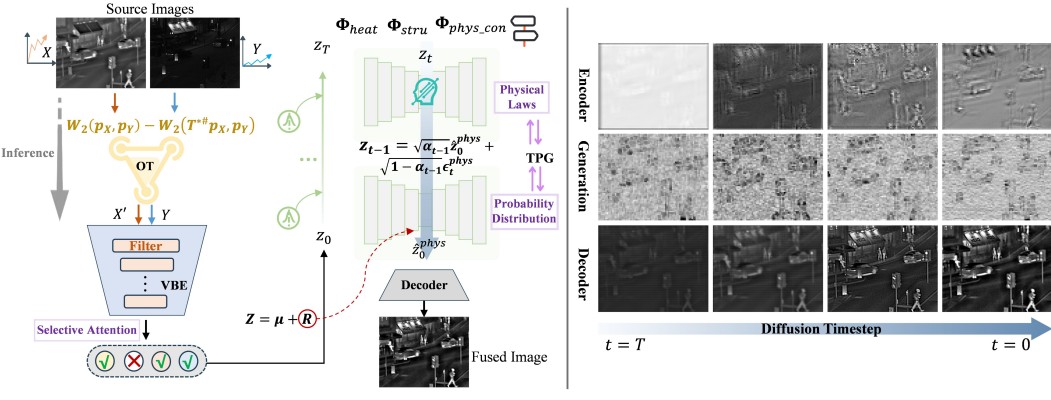

**(a) Overall framework of HCLFuse**  **(b) Feature Evolution Across Diffusion Timesteps**

Figure 2: Overall architecture of HCLFuse and feature evolution across the diffusion process.

architecture and optimization workflow of HCLFuse are provided in Appendix B. Ideally, the fused representation is expected to effectively preserve complementary information from both modalities while suppressing redundancy and noise. The image fusion task is formulated as a trade-off between compressive sensing and information preservation, aiming to optimize the selectivity of information to extract the most discriminative features. Specifically, assuming the fused representation follows a conditional distribution $Z \sim q(z|x,y)$, the optimization objective is defined as the maximization of the joint mutual information between $Z$ and the modal inputs:

$$\max_{q(z|x,y)} \mathcal{I}(Z; X, Y) \tag{1}$$

Here, the mutual information $\mathcal{I}(Z; X, Y)$ quantifies the amount of information from the modal inputs preserved in the fused representation[26], and is defined as follows:

$$\mathcal{I}(Z; X, Y) = \iint q(z, x, y) \log \frac{q(z|x,y)}{p(z)} \, dz \, dx \, dy \tag{2}$$

However, directly maximizing the information quantity may lead to the retention of excessive redundant content, which contradicts the selective attention mechanism observed in human cognitive systems. Within the predictive brain framework proposed by Clark[27], cognition is interpreted as an active process that adjusts internal models by minimizing prediction errors, thereby emphasizing the necessity of focusing on task-relevant information. To address this issue, the information bottleneck (IB)[28] theory is introduced as a mechanism to regulate information flow. The IB principle seeks a balance between compression and preservation by maximizing the mutual information between the fused representation and a task-relevant variable $C$, while minimizing its mutual information with the input:

$$\max_{q(z|x,y)} \mathcal{I}(Z; C) - \beta \mathcal{I}(Z; X, Y) \tag{3}$$

Here, $\beta$ serves as a trade-off coefficient that controls the balance between task relevance and compression. In unsupervised image fusion scenarios, explicit labels $C$ are unavailable. Therefore, a proxy task is designed by leveraging the modality alignment capability as a surrogate measure of task relevance.

**Theorem 1.** *(Lower Bound of Mutual Information under Unsupervised Mapping) Let modal inputs $X \sim p_X$ and $Y \sim p_Y$, with the fused representation $Z \sim q(z|x,y)$. Assume the existence of a latent task-relevant variable $C$ that satisfies the causal dependency $C \to (X, Y) \to Z$. Then, there exists an optimal transport mapping [29] $T^* : \mathcal{X} \to \mathcal{X}'$, such that the fused representation $Z$ generated from the transformed $X' = T^*(X)$ and $Y$ satisfies the following lower bound on mutual information (the full derivation is provided in Appendix A.1):*

$$\mathcal{I}(Z; C) \geq \mathcal{I}(Z; X', Y) - \varepsilon \geq \mathcal{I}(Z; X, Y) - \alpha \cdot \left[ W_2(p_X, p_Y) - W_2(T^{*\#} p_X, p_Y) \right] \tag{4}$$

where $\varepsilon > 0$ denotes an upper bound on the residual task-irrelevant information, $\alpha > 0$ is a sensitivity factor, $W_2(\cdot, \cdot)$ represents the second-order Wasserstein distance, and $T^{*\#} p_X$ denotes the pushforward distribution of $p_X$ under the optimal mapping $T^*$. This establishes a principled

foundation for fusion optimization through transport-based modality alignment. It reveals that the improvement in $\mathcal{I}(Z; C)$ is lower bounded by the reduction in Wasserstein distance after applying the optimal transport map $T^*$, offering a quantifiable and optimizable surrogate objective for information alignment under unsupervised conditions.

## 3.2 Variational Bottleneck Encoder

Under the guidance of the optimal transport mapping proposed in Theorem 1, a variational bottleneck encoder (VBE) is designed to extract salient and compact representations under information alignment. The input to the encoder consists of the concatenated transformed infrared image $X'$ and the visible image $Y$. Only the infrared image is transformed according to an optimal transport plan, while the visible image remains unchanged to ensure stability and efficiency. The transformation operator $T^*$ is obtained by multiplying the optimal transport plan $\mathbf{P}^*$ with the flattened infrared tensor $X_{\text{flat}} \in \mathbb{R}^{B \times N \times C}$, where $N = H \times W$:

$$T^*(X) = \mathbf{P}^* \cdot X_{\text{flat}}, \quad \mathbf{P}^* = \arg \min_{\mathbf{P} \in \mathcal{U}(r,c)} \sum_{i,j} P_{ij} C_{ij} + \varepsilon \sum_{i,j} P_{ij} \log P_{ij}. \tag{5}$$

Here, $\mathcal{U}(r, c)$ denotes the set of doubly stochastic matrices with row and column marginals $r$ and $c$, $C_{ij}$ represents the squared Euclidean distance between flattened pixel values of infrared and visible images, and $\varepsilon$ is a regularization coefficient. Through this transformation, the infrared image is geometrically and semantically aligned to the visible modality, reducing structural discrepancy between modalities and improving training efficiency. The transformed infrared image $X' = T^*(X)$ and the original visible image $Y$ are then jointly encoded to model their latent representation as $Z \sim q(Z|X', Y)$. The optimization objective of the VBE is formulated as:

$$\mathcal{L}_{\text{VBE}} = - \mathbb{E}_{q(Z|X',Y)}[\log p(Y|Z)] - \alpha \, \mathbb{E}_{q(Z|X',Y)}[\log p(X'|Z)]$$
$$+ \beta \, D_{\text{KL}}[q(Z|X', Y) \| p(Z)] \tag{6}$$

The first two terms evaluate the reconstruction capability of $Z$ with respect to $Y$ and $X'$, while the third term imposes a Kullback–Leibler (KL) divergence regularization to constrain the posterior $q(Z|X', Y)$ from deviating significantly from the prior $p(Z)$, thereby enabling controllable compression. The parameter $\beta > 0$ governs the strength of this information bottleneck constraint. The objective $\mathcal{L}_{\text{VBE}}$ essentially unifies the modeling principles of the Variational Autoencoder and the Information Bottleneck framework. It ensures robust cross-modal reconstruction while compressing redundant information in the latent space, thus generating structurally consistent and semantically compact fused representations $Z$ for the subsequent conditional modeling stage of the diffusion model. Under this bottleneck constraint, a multi-scale masking mechanism is further introduced to adaptively filter features at different scales, serving as a complementary enhancement rather than the bottleneck itself, thereby enhancing the expressiveness of the latent representation $Z$. This process is formulated as:

$$F_s = \text{concat}(X', Y), \quad F_m = \sigma \left( \theta_s \cdot (M_s \odot F_s) \right) \tag{7}$$

where $\sigma(\cdot)$ denotes the activation function, $\theta_s$ represents learnable parameters, and $M_s$ denotes the mask weights that determine the importance of each feature, which are learnable parameters obtained through a differentiable transformation $M_s = \text{sigmoid}(w_s)$, where $w_s \in \mathbb{R}^{1 \times C \times 1 \times 1}$ is initialized from a normal distribution and jointly optimized with $\mathcal{L}_{\text{VBE}}$. $F_s$ refers to the input of the VBE module. The operator $\odot$ indicates element-wise multiplication, through which the mask $M_s$ is applied to the features $F_s$ to obtain the condensed key feature representation $F_m$. To characterize the distributional properties of critical and uncertain information within the latent representation $Z$ more explicitly, the posterior distribution $q(Z|F_m)$ is modeled as a Gaussian distribution. Due to its continuity and differentiability, the Gaussian distribution facilitates sampling and optimization within the variational inference framework. Meanwhile, its parameterized structure enables effective modeling of both deterministic and uncertain components in the latent space. This modeling is expressed as:

$$q(Z|F_m) \sim \mathcal{N}(\mu, \sigma^2) \tag{8}$$

The mean $\mu$ and variance $\sigma^2$ of the latent variable $Z$ are computed from the masked features $F_m$ to represent the deterministic and uncertain components, respectively. This design enables a controllable generative capacity in the latent space while preserving discriminative semantic features from the multi-modal input, thereby facilitating improved expressiveness and structural consistency in the subsequent diffusion model. To further characterize the information structure within the latent

representation, the latent variable $Z \sim \mathcal{N}(\mu, \sigma^2)$ output by the variational bottleneck encoder is structurally decomposed. Specifically, considering that the encoder output contains both deterministic information driven by the input features and stochastic perturbations introduced by the variational modeling, the latent variable $Z$ can be expressed as:

$$Z = \mu + R, \quad R \sim \mathcal{N}(0, \sigma^2), \tag{9}$$

where $\mu$ denotes the mean vector computed from the masked features $F_m$, representing task-relevant structural information, and $R$ denotes a zero-mean Gaussian perturbation with covariance $\sigma^2$, modeling the uncertainty introduced during the information bottleneck compression process.

**Theorem 2** (Upper Bound of Redundant Mutual Information in the Perturbation Term). *Based on the decomposition in* (9)*, we consider the mutual information between the perturbation term $R$ and the task-relevant component $\mu$. Assuming a heteroscedastic Gaussian reparameterization consistent with the encoder implementation,*

$$R = \sigma \odot \varepsilon, \quad \varepsilon \sim \mathcal{N}(0, I),$$

*Under the joint-Gaussian and channel-diagonal dominance assumptions (see Appendix A.2), the redundant mutual information admits the following upper bound:*

$$\mathcal{I}(R; \mu) \leq \frac{1}{2} \sum_{i=1}^{d} \left[ -\log\left(1 - \frac{\mathrm{Var}[\mu_i]}{\sigma_i^2}\right) \right], \tag{10}$$

The latent variables decoupled by the variational bottleneck encoder provide critical features for the diffusion model. By constraining the redundancy in $R$, the diffusion process focuses on task-relevant structures, thereby improving the quality and consistency of the generated results. As shown in the first row of Fig. 2(b), the encoder output gradually captures refined and discriminative structural information, which is then fed into the subsequent generation and reconstruction processes.

## 3.3 Physics-Guided Conditional Diffusion Model

Diffusion models synthesize images via reverse denoising processes, capturing distributional patterns from large-scale data. Inspired by the interplay between empirical learning and physical reasoning in human cognition, a physics-guided conditional diffusion model is proposed, in which data-driven estimation is integrated with physically grounded constraints derived from domain knowledge. This hybrid mechanism enhances generalizability and interpretability while reducing dependence on high-quality data. In basic diffusion models, the reverse process (denoising sampling) is entirely based on the learned conditional distribution, typically formulated as:

$$p_\theta(z_{t-1} \mid z_t) \approx \mathcal{N}(\mu_\theta(z_t, t), \Sigma_\theta(z_t, t)) \tag{11}$$

To reinforce structural consistency and physical interpretability, the proposed physics-guided diffusion model introduces a physically grounded correction term $\Delta\mu_{\mathrm{phys}}(z_t, t)$ into the reverse process, with the VBE-generated latent $Z$ as input. The modified sampling is defined as:

$$p_\theta^{\mathrm{phys}}(z_{t-1} \mid z_t) \approx \mathcal{N}(\mu_\theta(z_t, t) + \Delta\mu_{\mathrm{phys}}(z_t, t), \Sigma_\theta(z_t, t)) \tag{12}$$

where $\Delta\mu_{\mathrm{phys}}(z_t, t)$ denotes a physics-based correction term that guides the generation process to obey fundamental physical laws. The probabilistic model provides an experience-driven initial estimate, while the physical constraints serve as rule-based corrections, jointly forming a generation process akin to the human cognitive laws between empirical experience and physical theorems. At each diffusion step $t$, generation proceeds in two stages: (1) a probabilistic estimate $\hat{z}_0$ is computed via denoising; (2) a physics-based correction $\hat{z}_0^{\mathrm{phys}} = \Phi_{\mathrm{physics}}(\hat{z}_0, t)$ is applied, followed by reverse sampling:

$$z_{t-1} = \sqrt{\alpha_{t-1}}\,\hat{z}_0^{\mathrm{phys}} + \sqrt{1 - \alpha_{t-1}}\,\epsilon_t^{\mathrm{phys}} \tag{13}$$

Where $\epsilon_t^{\mathrm{phys}}$ represents the re-estimated noise corresponding to $\hat{z}_0^{\mathrm{phys}}$, ensuring consistency in both semantic structure and noise components. This two-stage reasoning is executed at every timestep, enabling the generative trajectory to maintain data-driven capabilities while incorporating structural and physical corrections, leading to more realistic, stable, and physically plausible outputs. Under

the proposed physics-guided framework, three types of physical constraints are introduced to capture the intrinsic physical laws relevant to infrared and visible image fusion:

$$\Phi_{\text{physics}}(\hat{z}_0, t) = \Phi_{\text{con}}(\Phi_{\text{stru}}(\Phi_{\text{heat}}(\hat{z}_0, t), t), t) \tag{14}$$

**Heat Conduction Constraint.** This constraint reflects the law of energy transfer in physical materials, modeled on thermodynamic heat conduction. It describes the spatial diffusion of thermal energy across object surfaces, enforcing smooth and physically plausible energy distributions within homogeneous regions. The constraint is defined as:

$$\Phi_{\text{heat}}(\hat{z}_0, t) = \hat{z}_0 + \lambda_{\text{heat}}(t) \cdot \nabla^2 \hat{z}_0 \tag{15}$$

where $\nabla^2$ denotes the Laplacian operator, and $\lambda_{\text{heat}}(t)$ is the time-dependent heat diffusion coefficient. This constraint encourages the generated image to follow the heat conduction equation $\frac{\partial u}{\partial t} = \alpha \nabla^2 u$, thereby suppressing artifacts and discontinuities inconsistent with thermal physics.

**Structure Preservation Constraint.** Based on the assumption that object boundaries and structural features remain stable over short temporal intervals, this constraint is designed to maintain edge sharpness and shape consistency during fusion. It is formulated as:

$$\Phi_{\text{stru}}(\hat{z}_0^{\text{heat}}, t) = \hat{z}_0^{\text{heat}} + \lambda_{\text{stru}}(t) \left( G_{\text{max}} - G_{\hat{z}_0^{\text{heat}}} \right) M_{\text{stru}}, \tag{16}$$

Where $G_{\text{max}}$ denotes the maximum gradient map of the source image pair, providing structural edge information. $G_{\hat{z}_0^{\text{heat}}}$ is the gradient of the current estimate. And $M_{\text{stru}}$ is a structural mask derived from the high-frequency responses of the visible image, indicating important structural regions. This constraint enforces structural similarity to the source, ensuring the physical stability of prominent object boundaries.

**Physical Consistency Constraint.** This constraint enhances cross-modal physical coherence, ensuring that both modalities depict the object consistently in terms of physical properties. It is defined as:

$$\Phi_{\text{con}}(\hat{z}_0^{\text{stru}}, t) = \hat{z}_0^{\text{stru}} + \lambda_{\text{con}}(t) \left( w_{\text{ir}} \cdot X \cdot M_{\text{heat}} + w_{\text{vis}} \cdot Y \cdot M_{\text{stru}} \right), \tag{17}$$

Where $M_{\text{heat}}$ is the mask derived from the thermal–intensity distribution of the infrared image. Both masks are non-learnable spatial physical priors that guide the diffusion process toward physically plausible and cross-modally consistent regions. The weights $w_{\text{ir}}$ and $w_{\text{vis}}$ control the contributions of the infrared and visible modalities, respectively. To adapt to the varying uncertainty and mitigate potential bias introduced by imperfect masks during the diffusion process, a Time-varying Physical Guidance (TPG) mechanism is introduced, which defines each of the above $\lambda_i(t)$ as:

$$\lambda_i(t) = \lambda_i^0 \cdot e^{-\gamma t} \tag{18}$$

where $\lambda_i^0$ is the initial constraint weight, $\gamma$ is a decay factor, and $t$ is the normalized timestep. This mechanism reflects the cognitive process of "coarse perception followed by fine reasoning": in early steps (high noise, high uncertainty), stronger physical constraints provide guidance, while in later steps (low noise, high certainty), the model relies more on learned semantics and structural details for restoration. By leveraging this mechanism, the model dynamically adjusts physical guidance intensity, enabling robust and physically grounded cross-modal generation. As illustrated in Fig. 2(b), the visualized feature maps at different timesteps clearly exhibit this transition from noisy coarse perception to refined semantic restoration.

## 4  Experiments

### 4.1  Experimental Setup

HCLFuse is evaluated on four public datasets: MSRS [30], TNO [31], FMB [21] and MFNet [32], covering diverse conditions such as urban driving, nighttime military scenes, and adverse weather. Seventeen representative fusion methods are selected for comparison. Quantitative evaluations are performed using seven no-reference and five reference-based metrics. All experiments are implemented on an NVIDIA RTX 3090 GPU. Detailed descriptions of datasets, competing methods, hardware, and evaluation metrics are provided in Appendix C.1.

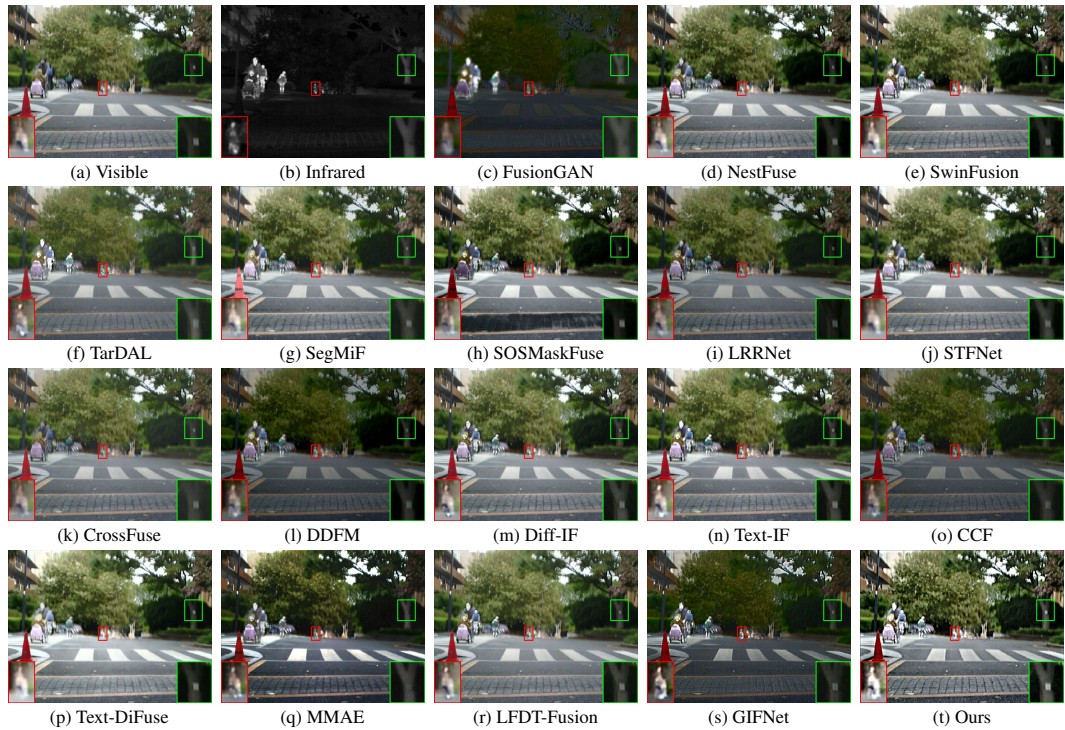

| (a) Visible | (b) Infrared | (c) FusionGAN | (d) NestFuse | (e) SwinFusion |
| (f) TarDAL | (g) SegMiF | (h) SOSMaskFuse | (i) LRRNet | (j) STFNet |
| (k) CrossFuse | (l) DDFM | (m) Diff-IF | (n) Text-IF | (o) CCF |
| (p) Text-DiFuse | (q) MMAE | (r) LFDT-Fusion | (s) GIFNet | (t) Ours |

Figure 3: Visualization results of several methods on MSRS dataset 00621D (image name) scene.

Table 1: The quantitative metrics of various algorithms in MSRS dataset. **Bold** indicates the best result. underline indicates the second-best result.

| Method | SD↑ | AG↑ | CC↑ | SCD↑ | EN↑ | SF↑ | Nabf↓ | DF↑ | QSF↑ | VIF↑ | PIQE↓ | BRI.↓ |
|---|---|---|---|---|---|---|---|---|---|---|---|---|
| FusionGAN | 19.644 | 1.6646 | 0.6276 | 1.0763 | 5.5537 | 5.0135 | 0.0239 | 1.9605 | -0.5557 | 0.4627 | 44.120 | 35.543 |
| NestFuse | 46.141 | 3.5573 | 0.5941 | 1.5632 | 6.1205 | 11.616 | 0.0112 | 4.0834 | -0.0207 | 0.9099 | 41.856 | 37.513 |
| SwinFusion | 47.651 | 3.7885 | 0.5900 | 1.5815 | 6.0590 | 12.550 | 0.0095 | 4.3502 | 0.0651 | 0.9119 | 38.137 | 37.551 |
| TarDAL | 35.460 | 3.1149 | 0.6261 | 1.4837 | 6.3476 | 9.8729 | 0.0098 | 3.8817 | -0.1419 | 0.6728 | 22.898 | **26.165** |
| SegMiF | 40.351 | 3.0449 | 0.6130 | 1.5482 | 6.3297 | 9.4273 | 0.0177 | 3.4624 | -0.1968 | 0.6239 | 42.081 | 34.445 |
| SOSMaskFuse | 45.647 | 3.3081 | 0.5492 | 1.2552 | 5.8463 | 11.248 | 0.0143 | 3.7733 | -0.0457 | 0.8585 | 44.769 | 40.064 |
| LRRNet | 36.849 | 3.0502 | 0.5171 | 0.8356 | 6.3341 | 9.8093 | 0.0208 | 3.6042 | -0.1617 | 0.5680 | 29.963 | 30.822 |
| STFNet | 46.980 | 3.3167 | 0.5992 | 1.5931 | 6.3750 | 9.9802 | 0.0113 | 3.6135 | -0.1389 | 0.8463 | 54.532 | 43.027 |
| CrossFuse | 36.309 | 3.0084 | 0.5433 | 1.0533 | 6.4947 | 9.6134 | 0.0243 | 3.5304 | -0.1805 | 0.8374 | 33.084 | 33.594 |
| DDFM | 28.923 | 2.5219 | **0.6585** | 1.4493 | 6.1748 | 7.3879 | 0.0196 | 2.9599 | -0.3581 | 0.74291 | 37.389 | 35.674 |
| Diff-IF | 42.598 | 3.7100 | 0.6023 | 1.6243 | 6.6686 | 11.460 | 0.0087 | 4.3206 | -0.0237 | **1.0417** | 32.734 | 31.515 |
| Text-IF | 44.588 | 3.8811 | 0.5982 | **1.6976** | 6.7280 | 11.879 | 0.0075 | 4.5075 | 0.0165 | **1.0506** | 33.987 | 31.681 |
| CCF | 28.946 | 2.8753 | 0.6495 | 1.4087 | 6.1917 | 9.0417 | 0.0154 | 3.6091 | -0.2369 | 0.6847 | **17.338** | 26.380 |
| Text-DiFuse | **54.243** | 3.7000 | 0.5710 | 1.3816 | **7.1436** | 11.408 | 0.0125 | 4.1898 | -0.0168 | 0.73116 | 35.283 | 34.988 |
| MMAE | 41.938 | 3.5325 | 0.6034 | 1.4173 | 6.1731 | 12.839 | 0.0087 | 4.1595 | 0.0719 | 0.8090 | 33.176 | 31.247 |
| LFDT-Fusion | 43.052 | 3.6429 | 0.6003 | 1.6370 | 6.6504 | 11.230 | 0.0095 | 4.1986 | -0.0422 | 1.0296 | 38.861 | 32.374 |
| GIFNet | 32.901 | 3.3673 | 0.6278 | 1.4082 | 5.9404 | 12.705 | 0.0110 | 3.8030 | 0.0654 | 0.5823 | 43.581 | 38.136 |
| Ours | 49.546 | **6.4355** | 0.6186 | 1.6575 | 6.8704 | **17.899** | **0.0017** | **7.6427** | **0.5374** | 0.8540 | 25.193 | 26.215 |

## 4.2 Qualitative Comparisons

Fig. 3 and Fig. 4 illustrate the fusion results of various methods on the MSRS dataset under daytime and nighttime scenes, respectively. As shown in Fig. 3, the red bounding box highlights pedestrian targets that are prominently captured in the infrared modality. Several methods, including FusionGAN, SOSMaskFuse, CrossFuse, and CCF, tend to over-enhance the thermal response, resulting in unnatural brightness distributions and noticeable visual artifacts. Additionally, MMAE fails to preserve the sign within the green box, leading to a critical loss of structural information—an example of severe fusion error. In contrast, only the proposed HCLFuse successfully preserves complementary features from both modalities, while also demonstrating a degree of detail restoration. For instance, fine details such as the bicycle wheels, leaves in the background, and pavement textures are clearly visible, indicating

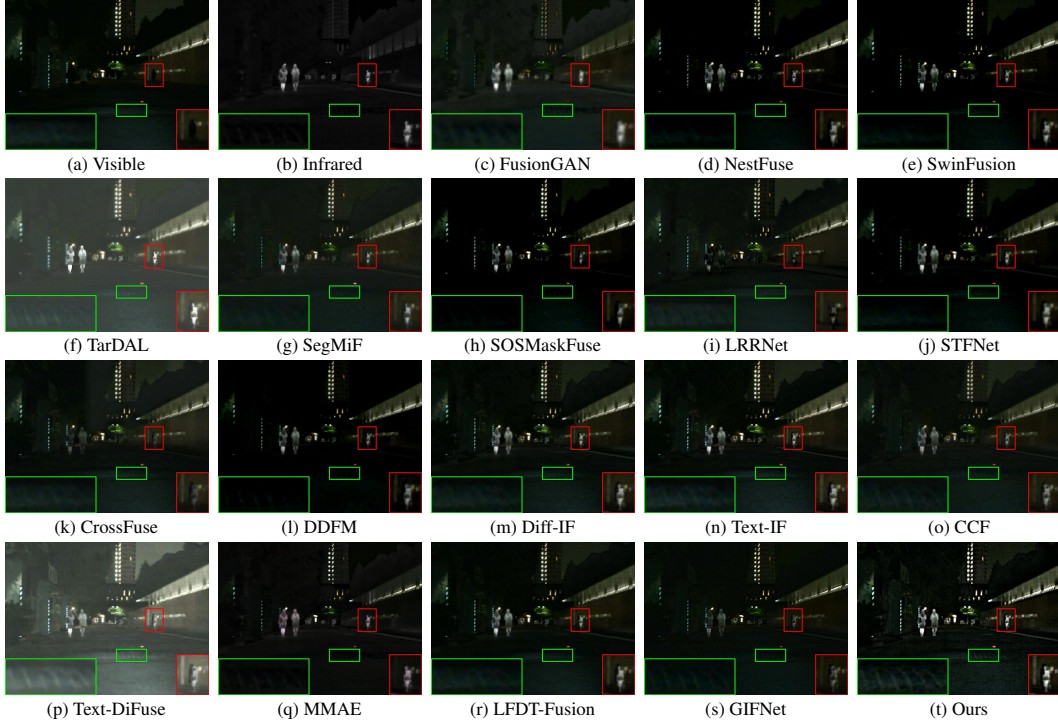

Figure 4: Visualization results of several methods on MSRS dataset 00774N scene.

better semantic preservation and structural coherence. As shown in Fig. 4 the scene is captured under low-light conditions, where infrared saliency becomes particularly important. However, methods such as LRRNet, CrossFuse and GIFNet fail to maintain the thermal prominence of pedestrians in the red box, thereby compromising target visibility. From a global perspective, TarDAL, Text-DiFuse, and HCLFuse preserve structural information in the green-box region, while only our method maintains higher perceptual resolution with clearer details. These results are generated by effectively integrating visible and infrared sources while maintaining structural integrity, thereby achieving superior perceptual quality without deviating from the underlying content.

## 4.3 Quantitative Comparisons

Table 1 reports the quantitative results of all compared methods on the MSRS dataset. The proposed HCLFuse achieves superior performance on most metrics, particularly excelling in texture clarity and structural fidelity. In terms of perceptual sharpness, HCLFuse obtains the highest AG, outperforming the second-best method by 69.87%, and reaches the best SF with a 39.41% relative gain, reflecting its strong capability in preserving fine-grained details. For DF, HCLFuse improves upon the next best result by 65.56%, indicating significantly enhanced visual clarity. Notably, HCLFuse achieves a substantially higher QSF score compared to all competing methods, demonstrating its superior capability in preserving directionally distributed frequency information. In addition, HCLFuse attains the highest EN, confirming its ability to maintain information richness while suppressing unnatural responses.

## 4.4 Generalization Evaluation

To further verify the robustness and generalization ability of HCLFuse across diverse datasets and scenarios, additional comparative experiments are conducted on the TNO, FMB, and MFNet datasets, with detailed quantitative and qualitative results presented in Appendix C, HCLFuse consistently outperforms existing fusion methods by leveraging its human cognition-inspired generative capability, while simultaneously maintaining strong generalization and robustness under varying conditions.

### 4.5 Downstream Task Evaluation

The ultimate goal of image fusion is to enhance the performance of downstream vision tasks. Among them, semantic segmentation places particularly strict demands on fine-grained semantic details. To evaluate this aspect, comparative experiments are conducted on the MSRS dataset using the Mask2Former[33] framework. As shown in Appendix C.6, HCLFuse achieves superior segmentation performance, attributed to its ability to retain fine structural and semantic cues, consistently outperforming other fusion baselines in this challenging downstream setting.

### 4.6 Ablation Studies

To evaluate the effectiveness of each component in HCLFuse, ablation experiments are conducted using the same quantitative metrics. As shown in Table 2, the complete model achieves the best performance on most indicators. In W/O TPG, the physics-guided constraint is removed, and sampling is performed purely through data-driven diffusion. While CC and Nabf show slight improvements, most metrics drop significantly, indicating unstable generation without physical priors. In W/O VBE, the VBE is replaced with a standard multi-scale encoder. Although this variant ranks second overall, visual artifacts such as coarse building textures and unnatural sky transitions appear (see Fig. 5), reflecting the model's reduced ability to filter and synthesize relevant features. In W/O OT, the optimal transport module proposed in Theorem 1 is removed, resulting in sharp declines across all metrics. This highlights the necessity of distribution alignment between modalities for stable fusion. In W/O DDIM, the deterministic diffusion sampling module (DDIM [34]) is disabled, which degrades both quantitative scores and visual quality. This confirms the critical role of the diffusion process in generating coherent fused images. More detailed ablation studies are presented in Appendix C.7 to further illustrate the effectiveness of the proposed method.

Table 2: Quantitative comparison of fusion performance in ablation studies on the effectiveness of designed modules. **Bold** indicates the best result. underline indicates the second-best result.

| | DDIM | OT | VBE | TPG | SD↑ | AG↑ | CC↑ | SCD↑ | EN↑ | SF↑ | Nabf↓ | DF↑ | QSF↑ | VIF↑ | PIQE↓ | BRI.↓ |
|---|---|---|---|---|---|---|---|---|---|---|---|---|---|---|---|---|
| W/O TPG | ✓ | ✓ | ✓ | ✗ | 36.90 | 5.521 | **0.646** | 1.595 | 6.495 | 15.158 | **0.0015** | 6.513 | 0.303 | 0.738 | **22.21** | 33.54 |
| W/O VBE | ✓ | ✓ | ✗ | ✗ | 42.68 | 6.038 | 0.607 | 1.521 | 6.736 | 17.259 | 0.0018 | **7.838** | 0.459 | 0.804 | 23.07 | 32.69 |
| W/O OT | ✓ | ✗ | ✗ | ✗ | 28.66 | 3.578 | 0.629 | 1.322 | 6.188 | 11.090 | 0.0107 | 4.391 | -0.058 | 0.734 | 23.97 | 29.85 |
| W/O DDIM | ✗ | ✗ | ✗ | ✗ | 28.36 | 3.626 | 0.635 | 1.337 | 6.176 | 11.218 | 0.0099 | 4.386 | -0.047 | 0.741 | 26.77 | 32.08 |
| Ours | ✓ | ✓ | ✓ | ✓ | **49.55** | **6.436** | 0.619 | **1.658** | **6.870** | **17.899** | 0.0017 | 7.643 | **0.537** | **0.854** | 25.19 | **26.22** |

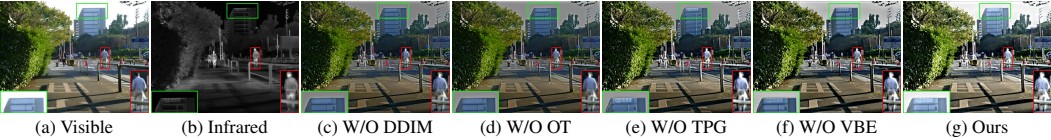

| (a) Visible | (b) Infrared | (c) W/O DDIM | (d) W/O OT | (e) W/O TPG | (f) W/O VBE | (g) Ours |
|---|---|---|---|---|---|---|

Figure 5: Visualization of ablation study results on the MSRS dataset.

## 5 Conclusion

A novel generative fusion framework is proposed by revisiting infrared and visible image fusion through the lens of human cognitive laws. Existing generative methods often lack modality interpretability and exhibit weak generative capability.To resolve these problems, a multi-scale mask-modulated variational bottleneck encoder grounded in information mapping theory is developed. This encoder enables accurate extraction of low-level modal cues, which significantly enhance structural fidelity during generation. Furthermore, physical laws are incorporated into the diffusion process to form a time-varying physical guidance mechanism, which enhances the model capacity to perceive intrinsic data structures and reduces dependence on data quality. HCLFuse achieves strong performance across various benchmarks. However, its reliance on well-aligned infrared and visible image pairs, together with the computational overhead introduced by the diffusion process, may limit its applicability in real-time or resource-constrained scenarios.

## Acknowledgement

This work was supported in part by the National Key Research and Development Program of China under Grant (2023YFF1105102, 2023YFF1105105), the National Natural Science Foundation of China under Grant 61772237, the Joint Fund of Ministry of Education for Equipment Pre-research under Grant 8091B042236.

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

# Appendix

## A  Proof

### A.1  Proof of Theorem 1

In the absence of explicit labels, the task-relevant variable $C$ is unobservable, which makes the direct optimization of $\mathcal{I}(Z;C) - \beta\,\mathcal{I}(Z;X,Y)$ intractable. We therefore derive a computable lower bound on $\mathcal{I}(Z;C)$ by aligning modal distributions via optimal transport.

#### A.1.1  Information Inequality from the Causal Structure

Given the causal Markov condition $C \to (X,Y) \to Z$, we have the conditional data-processing identity:

$$\mathcal{I}(Z;C) \;=\; \mathcal{I}(Z;X,Y) \;-\; \mathcal{I}(Z;X,Y \mid C) \;\geq\; \mathcal{I}(Z;X,Y) \;-\; \varepsilon \tag{19}$$

where $\varepsilon > 0$ is an upper bound on the residual task-irrelevant information conditioned on $C$. Let $T : \mathcal{X} \to \mathcal{X}'$ be a measurable map and set $X' = T(X)$. Since $X \to X' \to (Z \mid Y)$ is a Markov chain:

$$\mathcal{I}(Z;X',Y) \;\leq\; \mathcal{I}(Z;X,Y) \tag{20}$$

Combining (19) with (20) yields:

$$\mathcal{I}(Z;C) \;\geq\; \mathcal{I}(Z;X',Y) - \varepsilon \tag{21}$$

#### A.1.2  Effect of Distributional Transformation

View $\mathcal{I}(Z;X',Y) \;=\; H(Z) - H(Z \mid X',Y)$ as a functional of the pushed-forward marginal $p_{X'} = T^{\#}p_X$. Assume the conditional negative log-likelihood (decoder) $-\log q(z \mid x,y)$ is $L$-Lipschitz in $(x,y)$. By the Kantorovich–Rubinstein duality, there exists $L' > 0$ such that

$$\big| H(Z \mid X',Y) - H(Z \mid X,Y) \big| \;\leq\; L'\,W_1\big(p_X, T^{\#}p_X\big). \tag{22}$$

Using $W_1 \leq W_2$ yields

$$\big| H(Z \mid X',Y) - H(Z \mid X,Y) \big| \;\leq\; L'\,W_2\big(p_X, T^{\#}p_X\big). \tag{23}$$

Moreover, we consider $T$ chosen along the $W_2$ *displacement interpolation* from $p_X$ to $p_Y$ (i.e., $T = T_t$ with $t \in [0,1]$ on the McCann geodesic induced by the OT map), for which the metric projection satisfies

$$W_2\big(p_X, T^{\#}p_X\big) \;=\; \big| W_2(p_X,p_Y) \;-\; W_2\big(T^{\#}p_X,p_Y\big) \big|. \tag{24}$$

Combining (23) and (24), and absorbing constants into $\alpha > 0$, we obtain

$$\big| H(Z \mid X',Y) - H(Z \mid X,Y) \big| \;\leq\; \alpha\,\big| W_2\big(p_X,p_Y\big) - W_2\big(T^{\#}p_X,p_Y\big) \big|. \tag{25}$$

Hence, for any such $T$ along the geodesic we have

$$\mathcal{I}(Z;X',Y) \;\geq\; \mathcal{I}(Z;X,Y) \;-\; \alpha\,\big| W_2\big(p_X,p_Y\big) - W_2\big(T^{\#}p_X,p_Y\big) \big|. \tag{26}$$

#### A.1.3  Optimal Transport Map and Final Bound

Define the optimal transport map by:

$$T^* \;=\; \arg\min_{T}\; W_2\big(T^{\#}p_X,\, p_Y\big) \tag{27}$$

For $T = T^*$ we have $W_2(T^{*\#}p_X, p_Y) \leq W_2(p_X, p_Y)$, so the difference is non-negative and (26) gives:

$$\mathcal{I}(Z;X',Y) \;\geq\; \mathcal{I}(Z;X,Y) \;-\; \alpha\left[ W_2\big(p_X,p_Y\big) - W_2\big(T^{*\#}p_X,p_Y\big) \right] \quad \text{with } X' = T^*(X) \tag{28}$$

Finally, combining (21) and (28) yields the two-step chain in Theorem 1:

$$\mathcal{I}(Z;C) \;\geq\; \mathcal{I}(Z;X',Y) - \varepsilon \;\geq\; \mathcal{I}(Z;X,Y) \;-\; \alpha \cdot \left[ W_2\big(p_X,p_Y\big) - W_2\big(T^{*\#}p_X,p_Y\big) \right] \tag{29}$$

## A.2 Proof of Theorem 2

We consider $\mathcal{I}(R; \mu)$ with $R = z - \mu = \sigma \odot \varepsilon$ and $\varepsilon \sim \mathcal{N}(0, I)$. Let $\Sigma_\mu = \mathrm{Cov}(\mu)$, $\Sigma_R = \mathrm{Cov}(R)$ (diagonal, with entries $\sigma_i^2$ understood as the batch+spatial mean per channel), and $\Sigma_{R,\mu} = \mathrm{Cov}(R, \mu)$. Under the joint-Gaussian assumption, the mutual information admits the Schur-complement form:

$$\mathcal{I}(R; \mu) = \frac{1}{2} \log \frac{|\Sigma_R|}{|\Sigma_{R|\mu}|}, \qquad \Sigma_{R|\mu} = \Sigma_R - \Sigma_{R,\mu} \Sigma_\mu^{-1} \Sigma_{\mu,R}. \tag{30}$$

Define

$$M := \Sigma_R^{-1/2} \Sigma_{R,\mu} \Sigma_\mu^{-1} \Sigma_{\mu,R} \Sigma_R^{-1/2} \succeq 0. \tag{31}$$

Then (30) can be rewritten as

$$\mathcal{I}(R; \mu) = \frac{1}{2} \log \det\big((I - M)^{-1}\big) = -\frac{1}{2} \log \det(I - M). \tag{32}$$

To obtain a computable and conservative upper bound, we adopt a channel-diagonal dominance approximation,

$$M \preceq D := \mathrm{diag}(d_1, \ldots, d_d), \quad d_i := \frac{\big[\Sigma_{R,\mu} \Sigma_\mu^{-1} \Sigma_{\mu,R}\big]_{ii}}{\sigma_i^2} \leq \frac{\mathrm{Var}[\mu_i]}{\sigma_i^2}. \tag{33}$$

By Loewner order monotonicity, $(I - M)^{-1} \preceq (I - D)^{-1}$, hence

$$\det\big((I - M)^{-1}\big) \leq \det\big((I - D)^{-1}\big) = \prod_{i=1}^{d} \frac{1}{1 - d_i}. \tag{34}$$

Taking logarithm and using $d_i \leq \mathrm{Var}[\mu_i]/\sigma_i^2$ yields

$$\mathcal{I}(R; \mu) \leq \frac{1}{2} \sum_{i=1}^{d} \Big[ -\log(1 - d_i) \Big] \leq \frac{1}{2} \sum_{i=1}^{d} \Big[ -\log\Big(1 - \frac{\mathrm{Var}[\mu_i]}{\sigma_i^2}\Big) \Big], \tag{35}$$

# B  Algorithm

HCLFuse first applies an optimal-transport-based mapping $T^*$ to the infrared image $X$, aligning its distribution with that of the visible image $Y$ and thereby improving the optimization lower bound of the mutual-information objective. The aligned pair $(T^*(X), Y)$ is then fed into a multi-scale, mask-regulated variational bottleneck encoder (VBE) to compress and model the latent representation $z$, so that $z$ captures modality-discriminative and compact features under an unsupervised learning setting. Subsequently, $z$ is refined through a reverse-time diffusion generation process, in which physically guided constraints are dynamically injected at each denoising timestep to regulate the evolution of latent features. Finally, the optimized latent representation $z_0$ is decoded to produce the fused image $F$. The pseudocode implementations of both the training and inference procedures are provided in Algorithm 1 and Algorithm 2, respectively.

# C  Experimental Results

## C.1  Experimental Details

**Datasets.** To comprehensively assess the fusion performance of HCLFuse method, three publicly available datasets are utilized: MSRS[30], TNO[31], FMB [21], and MFNet [32]. The MSRS dataset provides 1,444 co-registered infrared and visible image pairs, primarily depicting urban driving scenes under both daytime and nighttime conditions. The TNO dataset contains 80 multispectral image pairs focused on nighttime military applications. FMB offers 1,500 aligned infrared-visible image pairs, covering a broad range of scenarios and illumination settings. Lastly, the MFNet dataset contains 1,569 pairs of co-registered RGB and thermal infrared images, captured in urban driving scenes under both daytime and nighttime conditions. In the experiments, a subset is sampled to ensure diversity and representative coverage: 361 pairs are selected from MSRS, 42 pairs from TNO, 280 pairs from FMB, and 393 pairs from MFNet. These selected subsets are used to validate the generalization capability of HCLFuse across varying scenes and lighting conditions.

---

**Algorithm 1** Training

---

**Input:** Source images $X$ and $Y$, total diffusion steps $T$
**Output:** Trained noise predictor $\epsilon_\theta$ and fused image $F$
 1: **for** $epoch = 1$ to $epochs$ **do**
 2:     $X' \leftarrow T^*(X)$
 3:     $z \leftarrow \text{VBE}(\text{concat}(X', Y))$
 4:     Sample $t \sim \text{Uniform}(\{1, \ldots, T\})$
 5:     Sample $\epsilon_t \sim \mathcal{N}(0, I)$
 6:     **for** $t = T, T-1, \ldots, 1$ **do**
 7:         $\epsilon_\theta \leftarrow \text{NoisePredictor}(z_t, z, t)$
 8:         Update $\lambda_i(t)$ using Eqs. (17–18)
 9:         Calculate the $z_0^{phys}$ using Eqs. (14–16)
10:         Calculate the $z_{t-1}$ using Eq. (13)
11:     $F \leftarrow \text{Decoder}(z_0^{phys})$
12:     Update $\epsilon_\theta$ using training loss

---

---

**Algorithm 2** Inference

---

**Input:** Source images $X$ and $Y$, total diffusion steps $T$
**Output:** Fused image $F$
 1: $z \leftarrow \text{VBE}(\text{concat}(X, Y))$
 2: Sample $z_T \sim \mathcal{N}(0, I)$
 3: **for** $t = T, T-1, \ldots, 1$ **do**
 4:     $\epsilon_\theta \leftarrow \text{NoisePredictor}(z_t, z, t)$
 5:     Update $\lambda_i(t)$ using Eqs. (17–18)
 6:     Calculate the $z_0^{phys}$ using Eqs. (14–16)
 7:     Calculate the $z_{t-1}$ using Eq. (13)
 8: $F = \text{Decoder}(z_0^{phys})$

---

**Competing Methods.** To comprehensively assess the effectiveness and robustness of HCLFuse method, comparisons are conducted against seventeen state-of-the-art image fusion methods. These include three non end-to-end methods: NestFuse[15], LRRNet[16], and MMAE[17]; seven end-to-end learning-based methods: SwinFusion[20], SegMiF[21], SOSMaskFuse[35], STFNet[22], CrossFuse[23], Text-IF[36] and GIFNet[37]; as well as seven generative approaches: FusionGAN[24], TarDAL[38], DDFM[39], Diff-IF[12], CCF[25], Text-DiFuse[40], and LFDT-Fusion[11]. All experimental evaluations are performed on a computational platform equipped with an NVIDIA GeForce RTX 3090 GPU and an Intel(R) Core(TM) i7-6850K CPU operating at 3.60 GHz.The Adam optimizer with a learning rate of $2 \times 10^{-5}$ is used for parameter updates.

**Metrics.** To quantitatively evaluate the fusion performance of HCLFuse, twelve metrics are adopted, consisting of seven no-reference indicators and five reference-based measures. The no-reference metrics include standard deviation (SD), average gradient (AG)[41], entropy (EN)[42], spatial frequency (SF)[43], definition (DF), perception-based image quality evaluator(PIQE)[44], and blind/referenceless image spatial quality evaluator (BRISQUE, abbreviated as BRI.)[45]. The reference-based metrics comprise the correlation coefficient (CC), the modified fusion artifacts measure (Nabf), the sum of correlations of differences (SCD)[46], the quality via spatial frequency (QSF)[47], and the visual information fidelity (VIF)[48].

## C.2 Comparison on TNO dataset

**Qualitative Evaluation.** Fig. 6 and Fig. 7 present visual comparisons between HCLFuse and 17 existing fusion methods on the TNO dataset. As a military-focused benchmark, TNO emphasizes the preservation of thermally salient targets under low-illumination conditions. In Fig. 6(c), (f), (i),(k), and (s), the thermal prominence of soldiers within the red box is noticeably suppressed by most baseline methods. In contrast, HCLFuse preserves high-contrast thermal features and structural detail, particularly in critical regions such as weapons and head contours, which appear more distinguishable from the background. In addition, surface textures—such as roof tiles—are reconstructed with enhanced clarity, indicating the generative capacity of the proposed diffusion-

based model in recovering fine-grained visual information. Similar superiority is observed in Fig. 7, where both target saliency and detail sharpness are consistently maintained across complex nighttime scenarios. Overall, the results produced by HCLFuse achieve a compelling balance between structural integrity and perceptual contrast, contributing to enhanced visual quality and improved target interpretability.

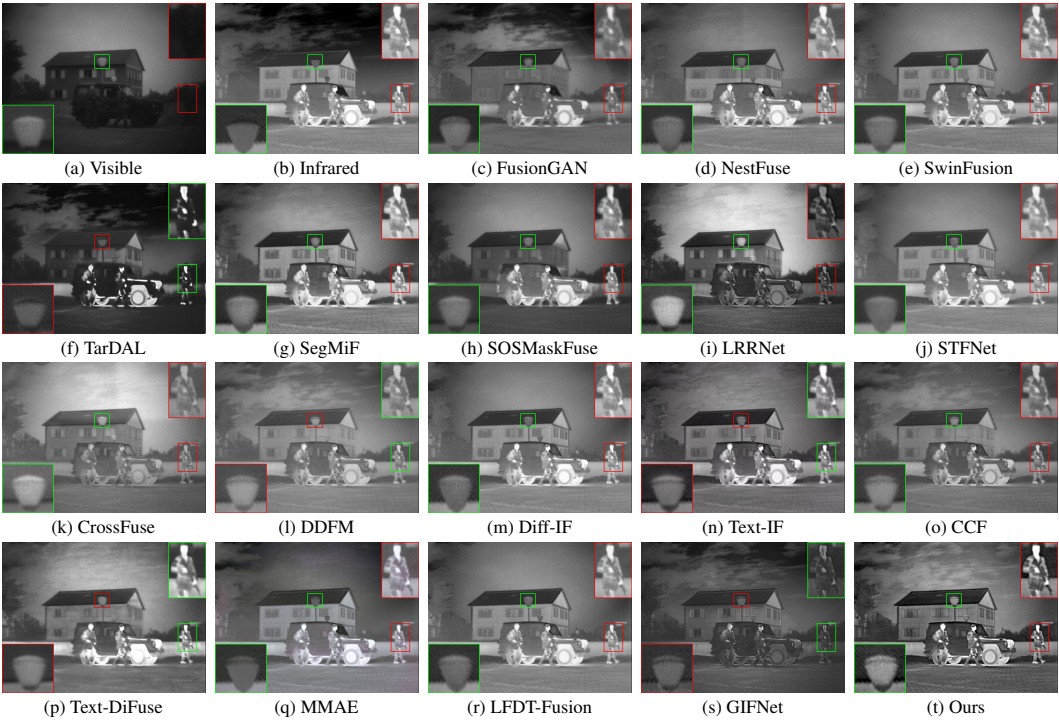

Figure 6: Visualization results of several methods on TNO dataset soldiers_with_jeep scene.

**Quantitative Evaluation.** As shown in Table 3, the proposed method consistently outperforms all competing approaches across most metrics. In particular, notable improvements are observed in the no-reference metrics, where several indicators exhibit substantial gains—for example, AG and DF improve by over 40% relative to the second-best results. The performance advantages demonstrated on the MSRS dataset are well preserved in the TNO dataset, highlighting the model's strong generalization capability. The observed robustness is attributed to the introduction of physics-guided sampling, which enhances the model's ability to capture the intrinsic structure of multimodal data. As a result, the fusion process becomes more stable and effective under varying scene conditions.

### C.3 Comparison on FMB dataset

**Qualitative Evaluation.** To further evaluate the adaptability of HCLFuse to complex and adverse environments, comparative experiments are conducted on the FMB dataset, which includes diverse weather conditions. The corresponding visual results are illustrated in Fig. 8 and Fig. 9. Fig. 8 shows a foggy daytime scene where the fusion objective lies in distinguishing salient targets from atmospheric interference. In this scenario, methods such as SegMiF, STFNet, CrossFuse, Diff-IF, Text-DiFuse, MMAE, and LFDT-Fusion fail to preserve the semantic integrity of the pedestrian target within the red box. Although other methods succeed in retaining this target, they struggle to reconstruct background structures, such as the high-rise building. Notably, FusionGAN discards almost all fog-related information, reflecting a biased fusion strategy favoring a single modality. In contrast, HCLFuse is capable of simultaneously preserving fog boundaries and target saliency while enhancing background texture fidelity. As a result, the output achieves a natural and balanced visual appearance. In Fig. 9, which presents a nighttime scenario, HCLFuse continues to emphasize global clarity and local saliency. Compared to other methods, it delivers a more comprehensive and

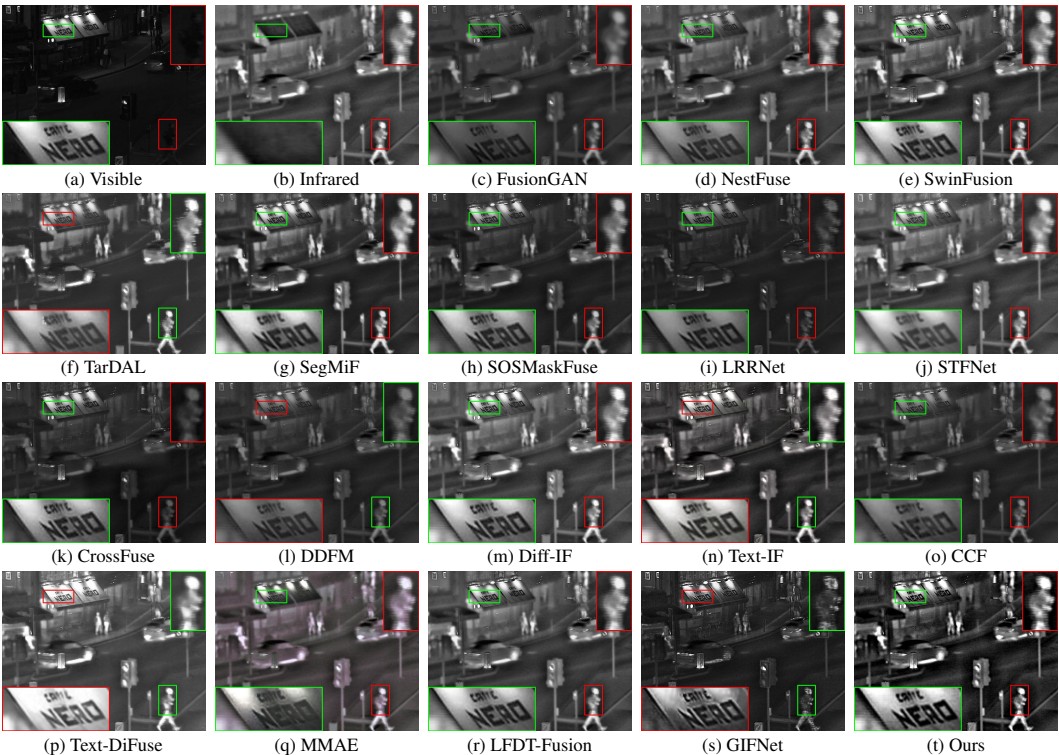

Figure 7: Visualization results of several methods on TNO dataset 042 scene.

Table 3: The quantitative metrics of various algorithms in TNO dataset. **Bold** indicates the best result. underline indicates the second-best result.

| Method | SD↑ | AG↑ | CC↑ | SCD↑ | EN↑ | SF↑ | Nabf↓ | DF↑ | QSF↑ | VIF↑ | PIQE↓ | BRI.↓ |
|---|---|---|---|---|---|---|---|---|---|---|---|---|
| FusionGAN | 30.663 | 2.4211 | 0.4404 | 1.3793 | 6.5580 | 6.2753 | 0.0816 | 3.2441 | -0.4550 | 0.4220 | 23.094 | 27.802 |
| NestFuse | 41.875 | 3.8485 | 0.4773 | 1.6899 | 7.0465 | 10.047 | 0.0328 | 4.9654 | -0.1315 | 0.8651 | 22.776 | 24.693 |
| SwinFusion | 39.447 | 4.2113 | 0.4744 | 1.7130 | 6.8909 | 10.722 | 0.0358 | 5.4839 | -0.1168 | 0.7503 | 20.655 | 24.113 |
| TarDAL | 40.141 | 3.8912 | 0.4538 | 1.5842 | 6.8079 | 10.621 | 0.0350 | 5.0487 | -0.0922 | 0.6006 | 21.454 | 24.665 |
| SegMiF | 47.609 | 4.2884 | 0.4657 | 1.6577 | 6.9097 | 10.721 | 0.0322 | 5.1762 | -0.0382 | 0.7028 | 23.350 | 25.405 |
| SOSMaskFuse | 44.896 | 3.8377 | 0.4264 | 1.5129 | 7.0393 | 10.161 | 0.0658 | 5.1514 | -0.1355 | 0.8765 | 21.640 | 25.888 |
| LRRNet | 40.879 | 3.7690 | 0.4461 | 1.5264 | 6.9881 | 9.5219 | 0.0557 | 5.0105 | -0.1674 | 0.5612 | 16.415 | 29.521 |
| STFNet | 37.997 | 2.8956 | 0.4467 | 1.5583 | 6.8148 | 6.9920 | 0.0497 | 3.3568 | -0.4066 | 0.7205 | 35.979 | 37.469 |
| CrossFuse | 39.674 | 3.7431 | 0.4015 | 1.3436 | 6.9075 | 9.9126 | 0.0731 | 5.1490 | -0.1427 | 0.7365 | 20.297 | 26.882 |
| DDFM | 34.295 | 3.3802 | **0.5307** | 1.7770 | 6.8496 | 8.5554 | 0.0443 | 4.3377 | -0.2628 | 0.6409 | **19.561** | 28.662 |
| Diff-IF | 39.245 | 4.2131 | 0.4468 | 1.5627 | 6.8949 | 11.344 | 0.0260 | 5.5671 | -0.0735 | 0.8433 | 20.635 | **22.832** |
| Text-IF | 46.866 | 4.6621 | 0.4614 | 1.6856 | **7.1878** | 11.752 | 0.0187 | 5.6968 | -0.0179 | 0.8110 | 27.487 | 32.399 |
| CCF | 36.888 | 2.8500 | 0.5227 | 1.7999 | 6.8925 | 7.3515 | 0.0512 | 3.4028 | -0.3740 | 0.5503 | 31.043 | 36.625 |
| Text-DiFuse | **51.276** | 3.0653 | 0.4516 | 1.6108 | 7.1521 | 8.0038 | 0.0602 | 3.4862 | -0.3407 | 0.4865 | 41.602 | 38.513 |
| MMAE | 39.987 | 3.5560 | 0.4292 | 1.5075 | 6.7764 | 10.151 | 0.0330 | 4.6762 | -0.1666 | 0.8229 | 25.222 | 26.164 |
| LFDT-Fusion | 40.100 | 4.1246 | 0.4483 | 1.5865 | 6.9395 | 10.892 | 0.0307 | 5.2708 | -0.1078 | **0.8774** | 23.424 | 25.183 |
| GIFNet | 40.406 | 4.9954 | 0.4966 | **1.8010** | 6.9213 | 13.358 | 0.0255 | 6.0598 | 0.0817 | 0.5045 | 22.094 | 34.594 |
| Ours | 47.726 | **7.2112** | 0.4838 | 1.7673 | 7.0975 | **17.625** | **0.0051** | **9.1041** | **0.5264** | 0.6137 | 25.710 | 29.259 |

perceptually coherent fusion result. This performance is attributed to the cognitive-guided fusion mechanism, where human perception principles are incorporated to enhance the model's ability to perceive, filter, and generate modality-specific information, resulting in high-quality generative fusion outputs.

**Quantitative Evaluation.** The quantitative results on the FMB dataset are reported in Table 4. It can be observed that the performance advantages previously demonstrated on the TNO and MSRS datasets are consistently maintained. Notably, each evaluation metric exhibits a considerable relative improvement over the second-best methods. Since the task of image fusion demands both high-fidelity generation and semantic consistency with the source modalities, the fusion results are expected to

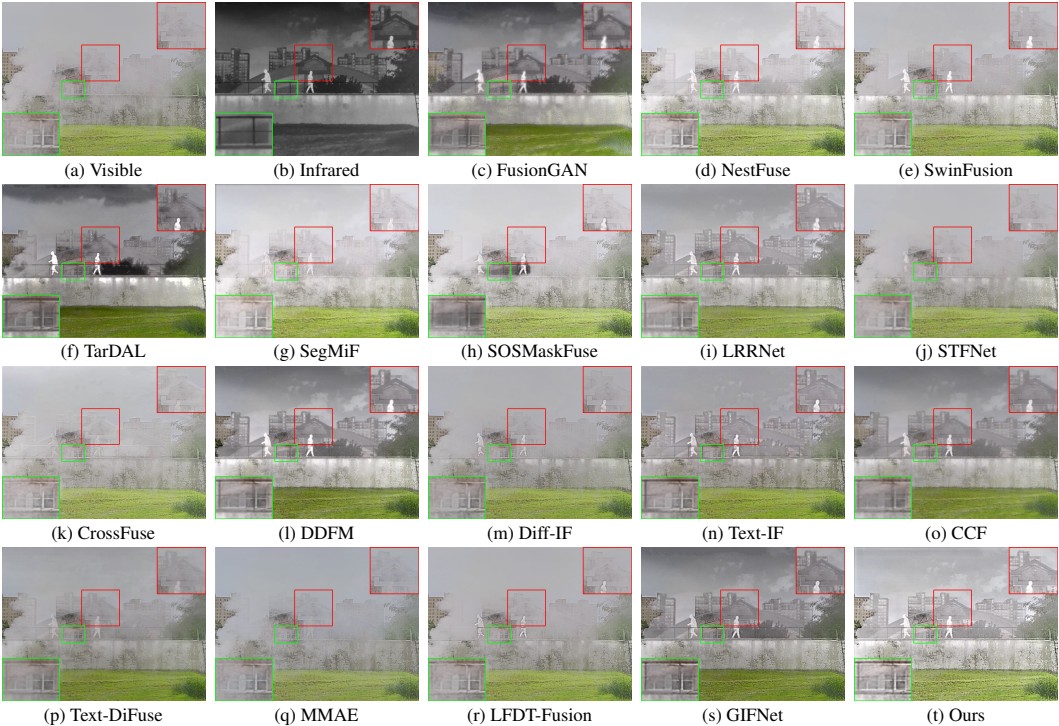

Figure 8: Visualization results of several methods on FMB dataset 00005 scene.

preserve the intrinsic characteristics of input images while enhancing perceptual quality. As evidenced across all experimental settings, HCLFuse effectively fulfills these dual objectives, demonstrating superior robustness and consistent performance gains across diverse conditions.

### C.4 Comparison on MFNet dataset

**Qualitative Evaluation.** To further investigate the robustness of HCLFuse, additional experiments are conducted on the MFNet dataset, which includes both daytime and nighttime scenarios. As illustrated in Fig. 10 and Fig. 11, most existing methods struggle to clearly reveal fine-grained details such as the bicycles in the background, often producing blurred and noisy regions. In contrast, HCLFuse achieves cleaner visual outputs through its pre-fusion information filtering mechanism, effectively suppressing irrelevant artifacts. Moreover, the diffusion-based generative process enhances the overall image quality and enriches structural and semantic information, yielding visually coherent and detailed fusion results.

**Quantitative Evaluation.** The quantitative results on the MFNet dataset are summarized in Table 5. Consistent with previous experiments, HCLFuse maintains stable superiority across all evaluation metrics. Notably, for the five leading indicators, the proposed method consistently ranks first across all four datasets, further highlighting the remarkable generative capability and robustness of HCLFuse in diverse fusion scenarios.

### C.5 Comparison on MFNet dataset

**Qualitative Evaluation.** To further investigate the robustness of HCLFuse, additional experiments are conducted on the MFNet dataset, which includes both daytime and nighttime scenarios. As illustrated in Fig. 10 and Fig. 11, most existing methods struggle to clearly reveal fine-grained details such as the bicycles in the background, often producing blurred and noisy regions. In contrast, HCLFuse achieves cleaner visual outputs through its pre-fusion information filtering mechanism, effectively suppressing irrelevant artifacts. Moreover, the diffusion-based generative process enhances the overall image quality and enriches structural and semantic information, yielding visually coherent and detailed fusion results.

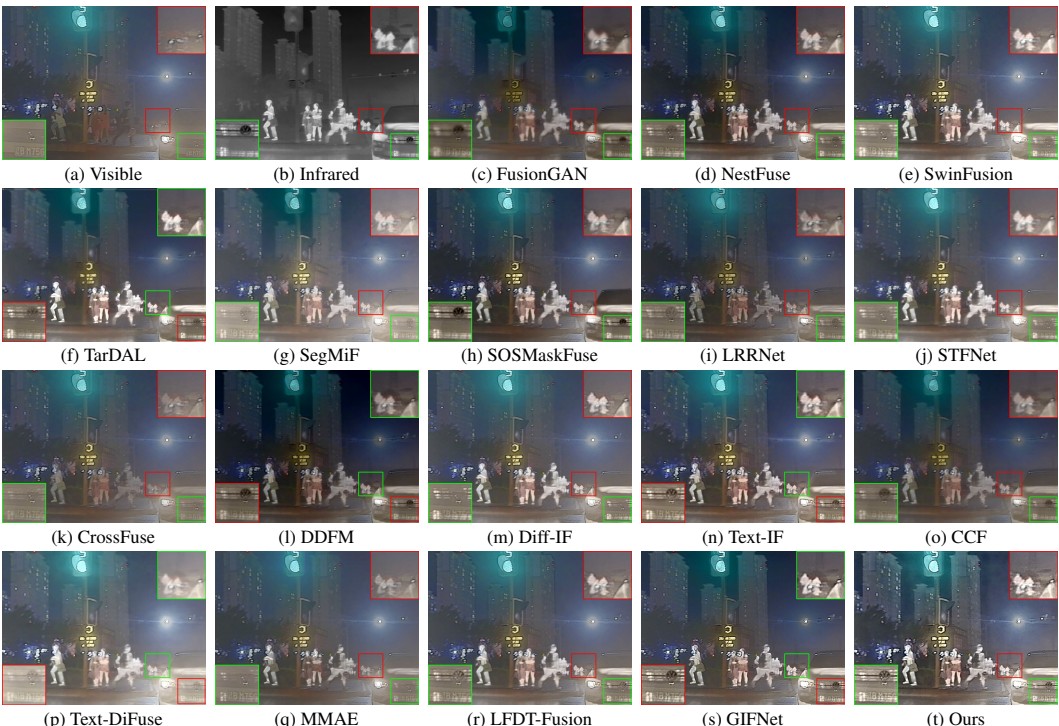

| | | | | | | | | | | | | | |
|---|---|---|---|---|---|---|---|---|---|---|---|---|---|
| (a) Visible | | (b) Infrared | | (c) FusionGAN | | (d) NestFuse | | (e) SwinFusion | | | | | |

Figure 9: Visualization results of several methods on FMB dataset 00001 scene.

Table 4: The quantitative metrics of various algorithms in FMB dataset. **Bold** indicates the best result. underline indicates the second-best result.

| Method | SD↑ | AG↑ | CC↑ | SCD↑ | EN↑ | SF↑ | Nabf↓ | DF↑ | QSF↑ | VIF↑ | PIQE↓ | BRI.↓ |
|---|---|---|---|---|---|---|---|---|---|---|---|---|
| FusionGAN | 33.175 | 2.8539 | 0.5637 | 1.2070 | 6.6987 | 9.6152 | 0.0152 | 3.5124 | -0.3202 | 0.4452 | 38.193 | 28.462 |
| NestFuse | 41.053 | 4.0541 | 0.6040 | 1.5107 | 6.8918 | 13.640 | 0.0070 | 4.8555 | -0.0352 | 0.8818 | 33.856 | 21.748 |
| SwinFusion | 40.766 | 4.6374 | 0.6194 | 1.6075 | 6.8552 | 15.496 | 0.0062 | 5.5574 | 0.0871 | 0.8965 | 30.361 | 24.851 |
| TarDAL | 40.943 | 3.3670 | 0.5873 | 1.5439 | 6.9166 | 11.196 | 0.0061 | 4.1873 | -0.2084 | 0.6338 | 28.446 | **18.729** |
| SegMiF | 38.091 | 3.7627 | 0.6076 | 1.5381 | 6.8625 | 11.766 | 0.0101 | 4.4029 | -0.1613 | 0.6722 | 37.482 | 22.024 |
| SOSMaskFuse | 36.870 | 4.3831 | 0.5464 | 1.1598 | 6.7717 | 14.866 | 0.0094 | 5.2575 | 0.0447 | **0.9628** | 32.527 | 22.598 |
| LRRNet | 30.471 | 3.5878 | 0.6223 | 1.3898 | 6.4809 | 11.814 | 0.0112 | 4.2965 | -0.1676 | 0.6324 | 33.792 | 19.414 |
| STFNet | 37.496 | 3.1591 | 0.5802 | 1.3915 | 6.7345 | 9.6157 | 0.0100 | 3.5607 | -0.3159 | 0.6476 | 51.921 | 30.633 |
| CrossFuse | 29.547 | 3.7301 | 0.5424 | 0.9238 | 6.4546 | 12.518 | 0.0147 | 4.5228 | -0.1135 | 0.8142 | 30.349 | 23.815 |
| DDFM | 31.975 | 2.7999 | **0.6615** | 1.6080 | 6.6920 | 9.0465 | 0.0119 | 3.3765 | -0.3608 | 0.6767 | 33.487 | 26.861 |
| Diff-IF | 34.229 | 4.0550 | 0.5837 | 1.3669 | 6.6349 | 13.870 | 0.0066 | 4.9305 | -0.0324 | 0.8717 | 29.142 | 20.140 |
| Text-IF | 34.552 | 4.5068 | 0.6031 | 1.5119 | 6.7451 | 15.050 | 0.0066 | 5.3928 | 0.0546 | 0.9518 | 30.216 | 22.977 |
| CCF | 37.310 | 2.2258 | 0.6476 | **1.7469** | 6.8266 | 7.4817 | 0.0135 | 2.5307 | -0.4742 | 0.5252 | 49.217 | 34.470 |
| Text-DiFuse | 39.806 | 3.3414 | 0.6121 | 1.5324 | 6.9141 | 11.475 | 0.0121 | 4.0719 | -0.1946 | 0.5996 | 26.496 | 37.312 |
| MMAE | 29.307 | 3.6648 | 0.5538 | 1.1878 | 6.4622 | 12.443 | 0.0109 | 4.3846 | -0.1285 | 0.8526 | 33.387 | 22.276 |
| LFDT-Fusion | 34.109 | 4.1304 | 0.5752 | 1.3411 | 6.6373 | 13.867 | 0.0082 | 4.9312 | -0.0292 | 0.7291 | 33.123 | 22.705 |
| GIFNet | 39.102 | 5.0481 | 0.6474 | 1.7283 | 6.9011 | 18.715 | 0.0043 | 5.9137 | 0.2999 | 0.5931 | 36.716 | 25.380 |
| Ours | **41.360** | **6.9814** | 0.6275 | 1.6321 | **7.0580** | **21.012** | **0.0018** | **8.8132** | **0.4947** | 0.7629 | **26.418** | 33.844 |

**Quantitative Evaluation.** The quantitative results on the MFNet dataset are summarized in Table 5. Consistent with previous experiments, HCLFuse maintains stable superiority across all evaluation metrics. Notably, for the five leading indicators, the proposed method consistently ranks first across all four datasets, further highlighting the remarkable generative capability and robustness of HCLFuse in diverse fusion scenarios.

## C.6 Segmentation comparison and analysis

The semantic segmentation performance is illustrated in Fig. 12 and Table 6. Fig. 12 presents visual comparisons on both daytime and nighttime scenes from the MSRS dataset. It can be observed that

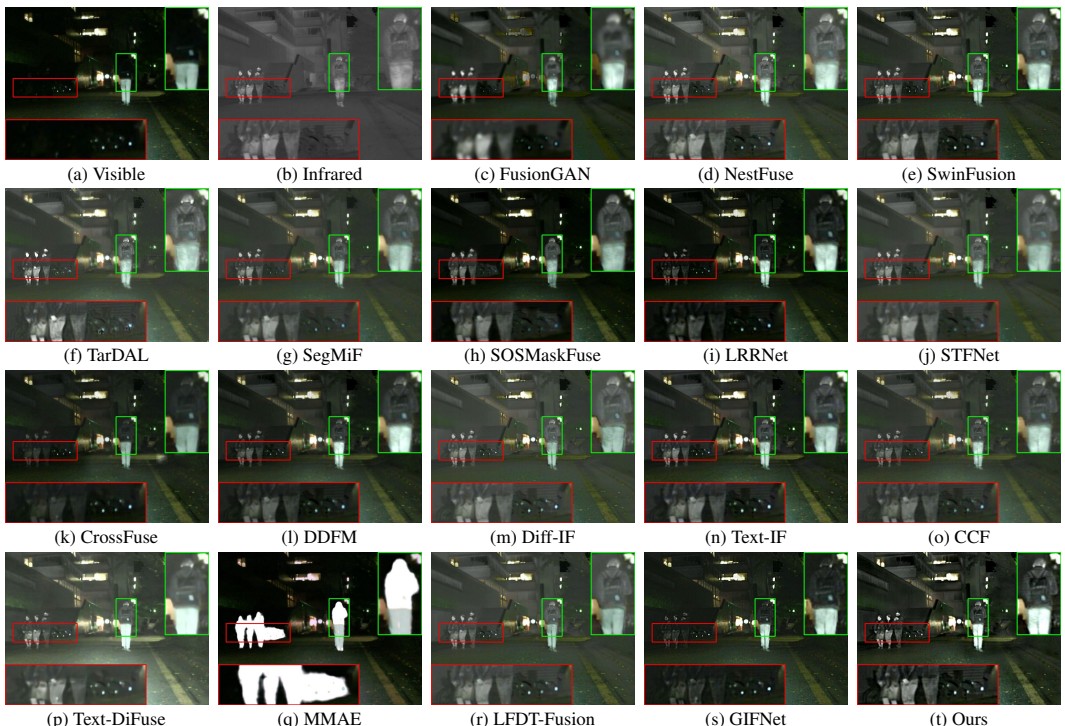

Figure 10: Visualization results of several methods on MFNet dataset 01304N scene.

Table 5: The quantitative metrics of various algorithms in MFNet dataset. **Bold** indicates the best result. underline indicates the second-best result.

| Method | SD↑ | AG↑ | CC↑ | SCD↑ | EN↑ | SF↑ | Nabf↓ | DF↑ | QSF↑ | VIF↑ | PIQE↓ | BRI.↓ |
|---|---|---|---|---|---|---|---|---|---|---|---|---|
| FusionGAN | 23.892 | 2.1156 | 0.5896 | 1.2885 | 6.0584 | 6.4544 | 0.0068 | 2.5495 | -0.4253 | 0.5563 | 38.692 | 33.244 |
| NestFuse | 39.939 | 3.6089 | 0.5677 | 1.5369 | 6.6525 | 11.437 | 0.0024 | 4.2067 | -0.0540 | 1.0281 | 38.603 | 33.102 |
| SwinFusion | 40.076 | 4.1712 | 0.5538 | 1.5156 | 6.5820 | 12.867 | 0.0024 | 4.8956 | 0.0816 | 0.9410 | 31.849 | 32.661 |
| TarDAL | 36.600 | 3.3083 | 0.5681 | 1.5599 | 6.6831 | 10.524 | 0.0035 | 4.0647 | -0.0536 | 0.7657 | 27.878 | **24.251** |
| SegMiF | 36.595 | 3.0624 | 0.5601 | 1.4905 | 6.3983 | 9.1332 | 0.0062 | 3.4995 | -0.2315 | 0.6699 | 44.895 | 34.382 |
| SOSMaskFuse | 43.790 | 3.7247 | 0.5087 | 1.3222 | 6.6309 | 11.868 | 0.0042 | 4.3045 | -0.0053 | 0.9405 | 40.792 | 35.119 |
| LRRNet | 32.355 | 2.9572 | 0.5356 | 1.3819 | 6.3787 | 9.1486 | 0.0066 | 3.4833 | -0.2174 | 0.7167 | 30.942 | 31.166 |
| STFNet | 36.427 | 3.0490 | 0.5554 | 1.4466 | 6.4178 | 9.0055 | 0.0033 | 3.3810 | -0.2399 | 0.8524 | 51.422 | 38.718 |
| CrossFuse | 34.719 | 3.2444 | 0.4836 | 1.1286 | 6.5573 | 9.9844 | 0.0081 | 3.8144 | -0.1568 | 0.8669 | 34.462 | 33.696 |
| DDFM | 30.751 | 2.7653 | **0.6082** | **1.6632** | 6.4816 | 8.2793 | 0.0051 | 3.2343 | -0.2938 | 0.7792 | 31.617 | 32.383 |
| Diff-IF | 33.823 | 3.5852 | 0.5467 | 1.3797 | 6.3762 | 11.468 | 0.0025 | 4.2272 | -0.0403 | 0.9191 | 36.260 | 33.043 |
| Text-IF | 41.337 | 4.0105 | 0.5394 | 1.5982 | 6.7777 | 12.275 | 0.0031 | 4.6470 | 0.0379 | **1.0711** | 36.734 | 32.907 |
| CCF | 28.369 | 2.7630 | 0.6077 | 1.5403 | 6.3183 | 8.9107 | 0.0041 | 3.4424 | -0.2492 | 0.7539 | **20.218** | 29.497 |
| Text-DiFuse | 48.269 | 3.5280 | 0.5262 | 1.4753 | **7.0612** | 10.971 | 0.0050 | 4.0200 | -0.0471 | 0.7792 | 36.220 | 34.261 |
| MMAE | **53.914** | 3.3900 | 0.3530 | 0.8330 | 6.5531 | 11.549 | 0.0042 | 3.8339 | 0.0505 | 0.8061 | 44.603 | 39.086 |
| LFDT-Fusion | 35.264 | 3.6937 | 0.5434 | 1.3966 | 6.4728 | 11.402 | 0.0033 | 4.2848 | -0.0428 | 0.9713 | 40.759 | 34.148 |
| GIFNet | 35.754 | 4.3341 | 0.5626 | 1.5210 | 6.3456 | 15.246 | 0.0035 | 4.9290 | 0.2699 | 0.6674 | 40.658 | 37.567 |
| Ours | 46.386 | **6.0428** | 0.5454 | 1.5466 | 6.9013 | **17.424** | **0.0007** | **7.2109** | **0.4782** | 0.9261 | 24.135 | 27.994 |

HCLFuse exhibits superior detail preservation and semantic awareness compared to other methods. In addition, the quantitative results in Table 6 show that HCLFuse consistently ranks among the top two across most metrics and achieves the highest mIoU score. These results demonstrate that the fused images generated by HCLFuse are more favorable for downstream semantic segmentation tasks, highlighting its notable advantage in semantic-level fusion quality.

## C.7 Additional ablation studies

**Effectiveness of physical constraints.** To validate the contribution of each physical constraint in HCLFuse, a comprehensive ablation study was conducted, as summarized in Table 7. Three

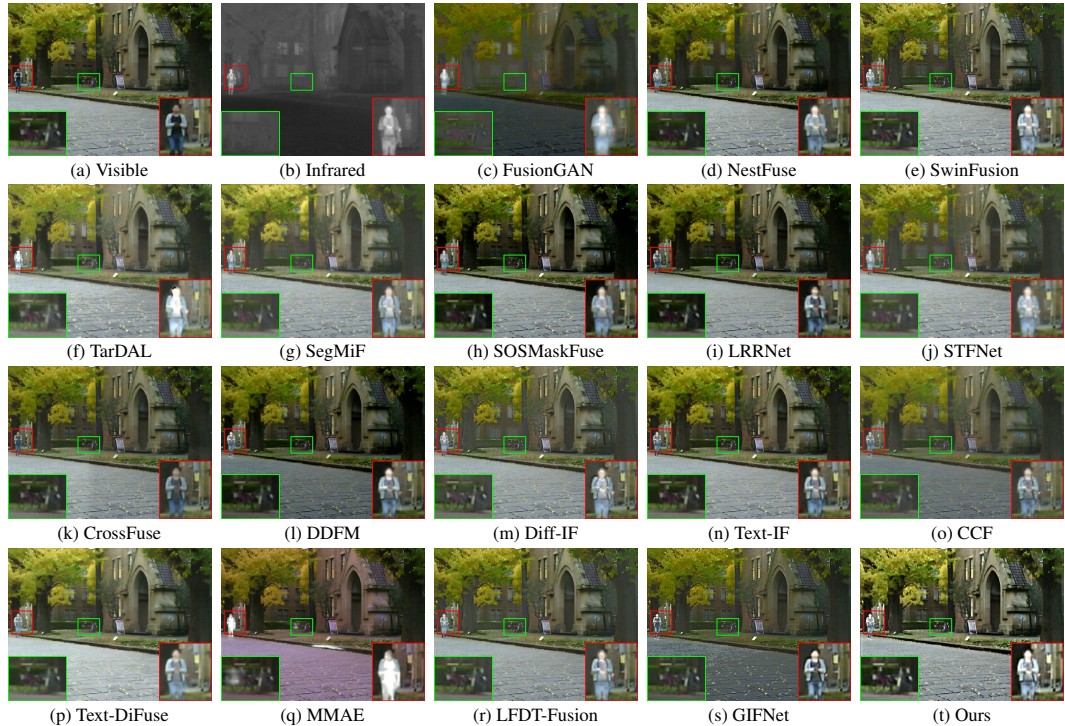

Figure 11: Visualization results of several methods on MFNet dataset 01493D scene.

Table 6: The quantitative metrics of various algorithms in semantic segmentation. **Bold** indicates the best result. underline indicates the second-best result.

| Method | unlabelled | car | person | bike | curve | car_stop | guardrail | color_cone | bump | mIOU |
|---|---|---|---|---|---|---|---|---|---|---|
| Visible | 98.97 | 93.86 | 75.50 | 85.48 | 76.63 | 85.46 | 91.56 | 77.04 | 88.82 | 85.92 |
| Infrared | 98.74 | 92.14 | 79.14 | 81.91 | 70.24 | 72.55 | 54.83 | 70.34 | 83.16 | 78.12 |
| FusionGAN | 99.05 | 93.71 | 81.89 | 85.60 | 77.54 | 83.25 | 89.74 | 75.76 | 86.34 | 85.88 |
| NestFuse | 99.11 | 94.13 | 82.78 | 86.08 | 78.04 | 85.68 | 92.62 | 76.77 | 89.53 | 87.19 |
| SwinFusion | 99.10 | 94.09 | 82.87 | 86.16 | 77.71 | 85.94 | 90.41 | 76.58 | 89.45 | 86.92 |
| TarDAL | 99.10 | 94.08 | 82.53 | 86.78 | 76.98 | 85.37 | 91.47 | 77.44 | 89.92 | 87.07 |
| SegMiF | 99.10 | 94.12 | 82.69 | 86.41 | 77.89 | 84.97 | 90.13 | 78.04 | 89.44 | 86.98 |
| SOSMaskFuse | 99.24 | 95.02 | 85.02 | 88.72 | 82.43 | 87.80 | 92.68 | 78.47 | 90.01 | 88.82 |
| LRRNet | 99.23 | 94.93 | 84.18 | 88.63 | 82.07 | 88.56 | 92.04 | 80.13 | 90.29 | 88.90 |
| STFNet | 99.23 | 94.99 | 84.83 | 88.83 | 82.21 | 87.47 | 92.51 | 80.49 | 90.64 | 89.02 |
| CrossFuse | 99.24 | 95.06 | 84.66 | 88.97 | 82.02 | 88.52 | 92.77 | 80.27 | 89.68 | 89.02 |
| DDFM | 99.22 | 95.05 | 84.68 | 88.48 | 81.95 | 87.39 | 92.06 | 78.27 | 89.72 | 88.54 |
| Diff-IF | 99.27 | 95.08 | **85.38** | **89.42** | 83.32 | 88.51 | 92.91 | **81.07** | 91.19 | 89.57 |
| Text-IF | 99.27 | 95.15 | 85.17 | 89.26 | 83.38 | 88.57 | 91.81 | 80.61 | **91.87** | 89.45 |
| CCF | 99.24 | 95.00 | 85.18 | 88.18 | 83.07 | 87.58 | 92.26 | 80.40 | 91.12 | 89.11 |
| Text-DiFuse | 99.27 | 95.07 | 84.6 | 89.24 | 84.18 | 88.52 | **93.08** | 80.87 | 91.55 | 89.60 |
| MMAE | 99.25 | 94.96 | 84.95 | 88.43 | 84.42 | 87.84 | 92.54 | 80.15 | 90.40 | 89.22 |
| LFDT-Fusion | **99.28** | **95.19** | 85.17 | 89.02 | 83.69 | 88.61 | 92.57 | 80.99 | 91.24 | 89.53 |
| GIFNet | 99.25 | 95.01 | 84.74 | 88.71 | 83.31 | 88.14 | 92.98 | 80.13 | 90.50 | 89.20 |
| Ours | **99.28** | 95.17 | 85.35 | 89.11 | **84.60** | **88.87** | 92.98 | 80.70 | 91.40 | **89.72** |

constraint terms were examined individually and jointly, including the *heat conduction constraint* ($\Phi_{\text{heat}}$), the *structure preservation constraint* ($\Phi_{\text{stru}}$), and the *physical consistency constraint* ($\Phi_{\text{con}}$). Introducing only $\Phi_{\text{heat}}$ leads to noticeable improvements in several perceptual indicators (e.g., AG, EN, and SF) and achieves the lowest Nabf and PIQE scores, indicating more stable and perceptually faithful generation. When $\Phi_{\text{stru}}$ is further incorporated, structural similarity metrics such as SCD are further enhanced, demonstrating that the structure constraint helps maintain edge integrity and sharpness. Finally, including the physical consistency term $\Phi_{\text{con}}$ yields the best overall performance

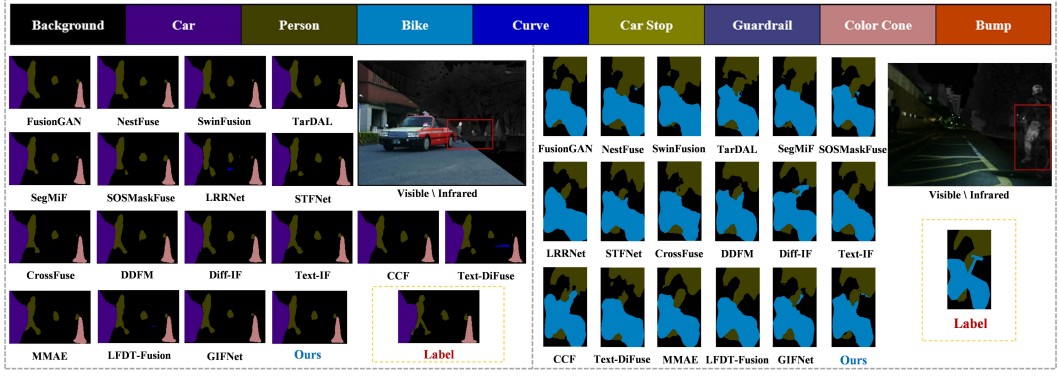

**(a) ' 00131D ' Scene**      **(b) ' 01262N ' Scene**

Figure 12: Segmentation visualization results of several methods based on the MSRS dataset.

across almost all metrics, with remarkable gains in SD, AG, EN, SF, and VIF. These results reveal the complementary and synergistic roles of the three physical constraints, collectively contributing to enhanced fidelity and stability of the generative process. Overall, the combination of all three physical terms produces the most balanced and high-quality fusion results, confirming that comprehensive physical modeling is essential for achieving optimal fusion performance.

Table 7: Ablation study on the effectiveness of the designed physical constraints. **Bold** indicates the best result.

| | $\Phi_{heat}$ | $\Phi_{stru}$ | $\Phi_{con}$ | SD↑ | AG↑ | CC↑ | SCD↑ | EN↑ | SF↑ | Nabf↓ | DF↑ | QSF↑ | VIF↑ | PIQE↓ | BRI.↓ |
|---|---|---|---|---|---|---|---|---|---|---|---|---|---|---|---|
| W/O $\Phi_{heat}$ | × | × | × | 43.11 | 6.615 | 0.503 | 1.809 | 6.929 | 15.95 | 0.004 | 8.126 | 0.384 | 0.520 | 20.37 | 37.25 |
| W/O $\Phi_{stru}$ | ✓ | × | × | 42.04 | 6.872 | 0.504 | 1.808 | 7.044 | 16.56 | **0.002** | 8.595 | 0.448 | 0.516 | **18.36** | 35.05 |
| W/O $\Phi_{con}$ | ✓ | ✓ | × | 41.25 | 6.705 | **0.508** | **1.818** | 7.025 | 16.51 | 0.003 | 8.694 | 0.432 | 0.505 | 21.21 | 35.35 |
| Ours | ✓ | ✓ | ✓ | **47.73** | **7.211** | 0.484 | 1.767 | **7.098** | **17.63** | 0.005 | **9.104** | **0.526** | **0.614** | 25.71 | **29.26** |

**Effectiveness of mask components.** To further investigate the effectiveness of the mask mechanisms in HCLFuse, we conducted a detailed ablation study covering all three types of masks involved in the framework. Two ablation settings were designed to assess their individual and combined contributions, as summarized in Table 8. W/O $M_s$ indicates that the semantic mask $M_s$ is removed from the latent representation, which leads to a clear performance degradation across multiple perceptual and structural metrics. The results confirm that $M_s$ plays an essential role in filtering informative latent variables and reducing redundancy within the bottleneck representation. In another setting, the heat and structure masks ($M_{heat}$ and $M_{stru}$) are replaced with all-one masks, effectively removing their spatial selectivity. Performance decreases consistently across key indicators related to detail and structure preservation (e.g., AG, SF, and QSF), indicating that omitting spatial mask guidance weakens both visual quality and perceptual fidelity. The quantitative results demonstrate that each mask plays an indispensable role within its respective mechanism: $M_s$ in the bottleneck pathway, and $M_{heat}$ and $M_{stru}$ in physically guided image generation. Collectively, these results highlight the necessity of multi-level mask modulation for achieving stable and high-quality fusion performance.

Table 8: Ablation study on the effectiveness of the mask components. **Bold** indicates the best result.

| | SD↑ | AG↑ | CC↑ | SCD↑ | EN↑ | SF↑ | Nabf↓ | DF↑ | QSF↑ | VIF↑ | PIQE↓ | BRI.↓ |
|---|---|---|---|---|---|---|---|---|---|---|---|---|
| W/O $M_s$ | 43.98 | 7.140 | 0.503 | 1.752 | **7.119** | 16.89 | **0.002** | 8.657 | 0.469 | 0.568 | 25.61 | 32.86 |
| W/O $M_{heat}$ & $M_{stru}$ | 39.16 | 6.821 | **0.507** | 1.749 | 6.991 | 15.74 | 0.003 | 7.974 | 0.369 | 0.517 | **19.12** | 39.70 |
| Ours | **47.73** | **7.211** | 0.484 | **1.767** | 7.098 | **17.63** | 0.005 | **9.104** | **0.526** | **0.614** | 25.71 | **29.26** |

## C.8 Broader impacts

Positive Impacts. The proposed method enhances image fusion quality and robustness in degraded scenarios, which can benefit applications such as autonomous driving, medical imaging, and disaster response by improving reliability and safety. Negative Impacts. Advanced fusion capabilities may

raise concerns about misuse in surveillance or military contexts. Additionally, the method involves computationally intensive models and assumes well-aligned inputs, which may limit accessibility and generalizability.

