# OpenReview forum: "Revisiting Generative Infrared and Visible Image Fusion Based on Human Cognitive Laws"
_NeurIPS.cc/2025/Conference — NeurIPS 2025 spotlight_

### Official Review · Reviewer_x9j9 · 2025-06-29

**Clarity:** 3
**Significance:** 3
**Originality:** 3
**Rating:** 5
**Confidence:** 5

**Summary:**

This paper introduces HCLFuse, a novel generative framework for infrared and visible image fusion, which is conceptually motivated by principles from human cognition.
It consists of two stage:
1. In the first stage, the authors design a Variational Bottleneck Encoder (VBE) to learn a compact yet comprehensive latent representation from the source images. The design and optimization of this encoder are formally grounded in the Information Bottleneck (IB) theory.
2. In the second stage, a conditional diffusion model synthesizes the final fused image from this latent code. The key innovation in this stage is that the reverse diffusion process is not purely data-driven. Instead, it is guided at each timestep by a set of proposed physics-based constraints, including terms that enforce heat conduction properties and structural preservation.

**Questions:**

I thank the authors for this novel and inspiring work. To further strengthen the paper and address the potential weaknesses, I hope the authors can provide further elaboration on the following key points. My final evaluation will be heavily influenced by the responses to the **weakness 1** and **weakness 2**.

**Ethical Concerns:**

["NO or VERY MINOR ethics concerns only"]

**Final Justification:**

I read all responses from the author. My concerns are mostly solved, and now I am learning to accept this paper.

**Limitations:**

Yes.

**Paper Formatting Concerns:**

N / A

**Quality:**

3

**Strengths And Weaknesses:**

## Strengths

1. The paper's primary strength is its exceptional novelty. It introduces a highly principled framework for image fusion, conceptually grounded in human cognition and physical laws. This represents a significant and refreshing departure from conventional methods that often rely on data-driven regression with generic losses (e.g., L1 or SSIM), opening a promising new direction for the field.
2. The paper is exceptionally well-written and logically structured. The authors present a compelling narrative that connects cognitive science, information theory, and physics-based modeling. Crucially, they support their claims with both detailed theoretical derivations (Theorems 1 and 2) and strong empirical evidence from extensive experiments.
3. The proposed method, HCLFuse, achieves state-of-the-art results. It demonstrates superior performance against a wide range of existing methods across multiple public datasets and evaluation metrics, in terms of both quantitative scores and qualitative visual quality.

## Weaknesses

1. The selection of baseline methods for comparison could be more contemporary and representative of the current state-of-the-art. A significant portion of the compared methods in Table 1 are from journals, which may have longer publication cycles and may not reflect the latest advancements. The comparison would be more compelling if the proposed method were benchmarked against more recent, highly relevant methods from top-tier conferences. For instance, recent works like TarDAL, DDFM, and the emerging diffusion-based or text-guided methods such as SegMIF, Text-IF, or Text-Difuse would serve as more challenging and appropriate baselines to truly validate the claimed state-of-the-art performance.
2. Insufficient Ablation of Individual Physical Constraints: In Section 3.3, the authors propose three distinct physical constraints: Heat Conduction, Structure Preservation, and Physical Consistency. However, the ablation study in Section 4.6 only evaluates their collective impact (the "w/o TPG" variant). This makes it impossible to understand the individual contribution of each constraint. The paper would be much stronger if it included a more fine-grained ablation study to demonstrate which of these physical principles are most critical to the model's success.
3. Ambiguity in Constraint Application: There is a minor ambiguity in the method's description in Section 3.3. Equation 14 applies the Heat Conduction constraint to the predicted clean data $\hat{z}_0$. However, the subsequent Equations 15 and 16, for the Structure and Consistency constraints, are formulated with the variable $z$. It should be clarified whether all constraints are applied to $\hat{z}_0$ or consistency, or if they are indeed applied to different $z_t$ during the whole recovery process.
4. Lack of Hyperparameter Analysis: The model introduces a significant number of hyperparameters, particularly the initial weights $\lambda_{i}^{0}$ and the decay factor $\gamma$ in the Time-varying Physical Guidance (TPG) mechanism. The paper does not provide details on how these were selected or discuss the model's sensitivity to them. A discussion of the tuning strategy and a sensitivity analysis would significantly improve the paper's claims of robustness and its practical value for reproducibility.

---

> ### Author Rebuttal · Authors · 2025-07-30
>
> We deeply appreciate your thoughtful and detailed review. In response, we have carefully revised the manuscript to address all your comments. A point-by-point response is provided below.
>
> **Response to Weakness 1:**
>
> We fully agree that comparing against more contemporary and competitive approaches is crucial to demonstrating the effectiveness of our proposed method. We have conducted extensive additional experiments to include the recent state-of-the-art methods mentioned in the review, such as TarDAL, DDFM, SegMIF, Text-IF, CCF, Text-DiFuse, and GIFNet. These methods represent leading advances from top-tier conferences. To present a more comprehensive comparison, we have  included a more challenging dataset, MFNet, to further validate the robustness and generalization ability of our method on low-quality image pairs.
> The additional experimental results are shown in the table below. To improve clarity, we use bold to indicate the best result and † to mark the second-best. Overall, the proposed method consistently achieves superior performance across most metrics, confirming its robustness and generalization ability even under comparison with the latest and strongest baselines.
>
> （Due to the rebuttal's character limit, we only present the fusion performance here, while the evaluation on the segmentation task is included in the revised manuscript. If needed, we would be happy to provide those results separately.）
>
> **(1)MSRS Dataset**
> | Metric     | TARDAL (CVPR 2022) | Seg_MiF (ICCV 2023) | DDFM (ICCV 2023) | Text-IF (CVPR 2024) | CCF (NeurIPS 2024) | Text-DiFuse (NeurIPS 2024) | GIFNet (CVPR 2025) | Ours |
> |:-----|:---------------|:----------------|:-------------|:----------------|:---------------|:------------------------|:----------------|:----|
> |SD↑|35.4604|40.3512|28.9225|44.5884|28.9463|**54.2426**|32.9010|49.5464†|
> |AG↑|3.1149|3.0449|2.5219|3.8811†|2.8753|3.7000|3.3673|**6.4355**|
> |CC↑|0.6261|0.6130|**0.6585**|0.5982|0.6495†|0.5710|0.6278|0.6186|
> |SCD↑|1.4837|1.5482|1.4493|**1.6976**|1.4087|1.3816|1.4082|1.6575†|
> |EN↑|6.3476|6.3297|6.1748|6.7280|6.1917|**7.1436**|5.9404|6.8704†|
> |SF↑|9.8729|9.4273|7.3879|11.8790|9.0417|11.4080|12.7048†|**17.8985**|
> |Nabf↓|0.0098|0.0177|0.0196|0.0075†|0.0154|0.0125|0.0110|**0.0017**|
> |DF↑|3.8817|3.4624|2.9599|4.5075†|3.6091|4.1898|3.8030|**7.6427**|
> |QSF↑|-0.1419|-0.1968|-0.3581|0.0165|-0.2369|-0.0168|0.0654†|**0.5374**|
>
> **(2)FMB Dataset**
>
> |Metric|TARDAL (CVPR 2022)|Seg_MiF (ICCV 2023)|DDFM (ICCV 2023)|Text-IF (CVPR 2024)|CCF (NeurIPS 2024)|Text-DiFuse (NeurIPS 2024)|GIFNet (CVPR 2025)|Ours|
> |:-----|:----------------|:------------------|:---------------|:------------------|:------------------|:--------------------------|:------------------|:----|
> |SD↑|40.9429†|38.0909|31.9747|34.5523|37.3103|39.8064|39.1017|**41.3603**|
> |AG↑|3.3670|3.7627|2.7999|4.5068|2.2258|3.3414|5.0481†|**6.9814**|
> |CC↑|0.5873|0.6076|**0.6615**|0.6031|0.6476†|0.6121|0.6474|0.6275|
> |SCD↑|1.5439|1.5381|1.6080|1.5119|**1.7469**|1.5324|1.7283†|1.6321|
> |EN↑|6.9166†|6.8625|6.6920|6.7451|6.8266|6.9141|6.9011|**7.0580**|
> |SF↑|11.1957|11.7657|9.0465|15.0501|7.4817|11.4746|18.7147†|**21.0123**|
> |Nabf↓|0.0061|0.0101|0.0119|0.0066|0.0135|0.0121|0.0043†|**0.0018**|
> |DF↑|4.1873|4.4029|3.3765|5.3928|2.5307|4.0719|5.9137†|**8.8132**|
> |QSF↑|-0.2084|-0.1613|-0.3608|0.0546|-0.4742|-0.1946|0.2999†|**0.4947**|
>
> **(3)TNO Dataset**
>
> |Metric|TARDAL (CVPR 2022)|Seg_MiF (ICCV 2023)|DDFM (ICCV 2023)|Text-IF (CVPR 2024)|CCF (NeurIPS 2024)|Text-DiFuse (NeurIPS 2024)|GIFNet (CVPR 2025)|Ours|
> |:-----|:----------------|:------------------|:---------------|:------------------|:------------------|:--------------------------|:------------------|:----|
> |SD↑|40.1405|47.6092|34.2954|46.8663|36.8877|**51.2755**|40.4060|47.7260†|
> |AG↑|3.8912|4.2884|3.3802|4.6621|2.8500|3.0653|4.9954†|**7.2112**|
> |CC↑|0.4538|0.4657|**0.5307**|0.4614|0.5227†|0.4516|0.4966|0.4838|
> |SCD↑|1.5842|1.6577|1.7770|1.6856|1.7999†|1.6108|**1.8010**|1.7673|
> |EN↑|6.8079|6.9097|6.8496|**7.1878**|6.8925|7.1521†|6.9213|7.0975|
> |SF↑|10.6209|10.7205|8.5554|11.7515|7.3515|8.0038|13.3580†|**17.6254**|
> |Nabf↓|0.0350|0.0317|0.0443|0.0187†|0.0512|0.0602|0.0255|**0.0051**|
> |DF↑|5.0487|5.1762|4.3377|5.6968|3.4028|3.4862|6.0598†|**9.1041**|
> |QSF↑|-0.0922|-0.0382|-0.2628|-0.0179|-0.3740|-0.3407|0.0817†|**0.5264**|
>
>
> **(4)MFNet Dataset**
>
> | Metric   | TARDAL (CVPR 2022) | Seg_MiF (ICCV 2023) | DDFM (ICCV 2023) | Text-IF (CVPR 2024) | CCF (NeurIPS 2024) | Text-DiFuse (NeurIPS 2024) | GIFNet (CVPR 2025) | Ours |
> |:---------|:-------------------|----------------------|------------------|----------------------|--------------------|-----------------------------|--------------------|------|
> |SD↑|36.6004|36.5946|30.7512|41.3369|28.3691|**48.2688**|35.7542|46.3864†|
> |AG↑|3.3083|3.0624|2.7653|4.0105|2.7630|3.5280|4.3341†|**6.0428**|
> |CC↑|0.5681|0.5601|**0.6082**|0.5394|0.6077†|0.5262|0.5626|0.5454|
> |SCD↑|1.5599|1.4905|**1.6632**|1.5982†|1.5403|1.4753|1.5210|1.5466|
> |EN↑|6.6831|6.3983|6.4816|6.7777|6.3183|**7.0612**|6.3456|6.9013†|
> |SF↑|10.5243|9.1332|8.2793|12.2754|8.9107|10.9706|15.2457†|**17.4240**|
> |Nabf↓|0.0035|0.0062|0.0051|0.0031†|0.0041|0.0050|0.0035|**0.0007**|
> |DF↑|4.0647|3.4995|3.2343|4.6470|3.4424|4.0200|4.9290†|**7.2109**|
> |QSF↑|-0.0536|-0.2315|-0.2938|0.0379|-0.2492|-0.0471|0.2699†|**0.4782**|
>
> **Response to Weakness 2:**
>
> We have added a new ablation study that systematically isolates and analyzes each individual constraint. As shown in the table below, the progression from the baseline (without any physical guidance) to successively adding these constraints yields clear and interpretable results:
>
> From A → B, introducing only $\Phi_{\text{heat}}$ significantly boosts perceptual indicators such as AG, EN, and SF, while also achieving the lowest Nabf. This demonstrates that enforcing thermal smoothness leads to tangible benefits even in isolation.
>
> From B → C, adding $\Phi_{\text{stru}}$ leads to further improvements in structure-related metrics such as SCD, confirming its effectiveness in preserving edges and local details.
>
> From C → Ours, incorporating the full $\Phi_{\text{con}}$ term further enhances global consistency and cross-modal balance, resulting in the best overall performance across nearly all metrics, including SD, AG, VIF, and QSF.
>
> This layered ablation demonstrates that each constraint contributes uniquely to different aspects of fusion quality. More importantly, the full combination of all three constraints achieves the most balanced and robust results.
>
> The best results are highlighted in bold.
>
> ||$\Phi_\text{heat}$|$\Phi_\text{stru}$|$\Phi_\text{con}$|SD↑|AG↑|CC↑|SCD↑|EN↑|SF↑|Nabf↓|DF↑|QSF↑|
> |----------------------------|------|------|------|---------|---------|---------|---------|---------|---------|---------|---------|---------|
> |A|✗|✗|✗|43.1107|6.6147|0.5028|1.8086|6.9285|15.9481|0.0040|8.1256|0.3838|
> |B|✓|✗| ✗|42.0447|6.8718|0.5035|1.8076|7.0436|16.5597|**0.0029**|**18.3564**|35.0540|
> |C|✓|✓| ✗|41.2477|6.7049|**0.5080**|**1.8183**|7.0248|16.5085|0.0034|8.6943| 0.4319|
> |Ours|✓|✓|✓|**47.7260**|**7.2112**|0.4838|1.7673|**7.0975**|**17.6254**| 0.0051|**9.1041**|**0.5264** |
>
> **Response to Weakness 3:**
>
> We appreciate your comment, which helped us correct this oversight. The descriptions of the three physical constraints in Equations (14)–(16) are indeed inaccurate. In fact, these constraints are applied sequentially in a cascaded manner. We have revised the manuscript to clearly state that the constraints are applied in order at each timestep:
>
> z0_hat --> $\Phi_{\text{heat}}$ --> z0_hat^heat --> $\Phi_{\text{stru}}$ --> z0_hat^stru --> $\Phi_{\text{con}}$ --> z0_hat^phys
>
> **Response to Weakness 4:**
>
> In response to this suggestion, we have conducted a comprehensive study on the key hyperparameters used in our framework. Specifically, we divided them into three groups and performed targeted experiments to evaluate their influence on performance.
>
> (1) Analysis of $\beta$ in Equation (5):
> We evaluated the trade-off coefficient $\beta$ (values: 0.1, 1, 10), which balances compression and information preservation in the variational bottleneck. The best performance occurred at $\beta = 1$, achieving an effective trade-off between KL regularization and reconstruction. Smaller $\beta$ led to under-regularization and instability, while larger values suppressed informative features.
>
> (2) Analysis of the physical guidance weights $\lambda_{\text{heat}}$, $\lambda_{\text{stru}}$, and $\lambda_{\text{con}}$ in Equations (14–16):
> A grid search over values {0.05, 0.10, 0.15} for each $\lambda$ (with others fixed) yielded six combinations. The best result is with $(\lambda_{\text{heat}} = 0.05, \lambda_{\text{stru}} = 0.15, \lambda_{\text{con}} = 0.10)$, reflecting an effective balance: low $\lambda_{\text{heat}}$ prevents over-smoothing, high $\lambda_{\text{stru}}$ preserves structure, and moderate $\lambda_{\text{con}}$ promotes stable cross-modal fusion.
>
> (3) Analysis of the decay factor $\gamma$ in Equation (17):
> We tested $\gamma = 1, 5, 10$ to examine its effect on time-dependent physical guidance. The optimal $\gamma = 5$ maintained sufficient guidance while preserving flexibility. $\gamma = 1$ was too weak, causing blurred structures; $\gamma = 10$ was too strong, degrading performance by over-constraining the model.
>
> Due to the rebuttal's character limit, only a brief summary of the quantitative results is included here. And the complete ablation results are presented in the revised manuscript. If you would like to review the complete results, we would be happy to provide them in a follow-up.
>
> **Response to Question:**
>
> We have sincerely and thoroughly addressed all the concerns you raised. The manuscript has been carefully revised based on your suggestions, and we look forward to your further feedback.

---

> > ### Comment · Reviewer_x9j9 · 2025-08-04
> >
> > Thank you very much for your detailed response. My concerns are mostly solved and now I learn towards accepting this paper.

---

> > > ### Author Response · Authors · 2025-08-04
> > >
> > > Thank you very much for your encouraging feedback. We are grateful for your thoughtful review and are pleased that our clarifications have addressed your concerns.

---

### Official Review · Reviewer_cS6j · 2025-06-30

**Clarity:** 3
**Significance:** 3
**Originality:** 3
**Rating:** 4
**Confidence:** 5

**Summary:**

This paper proposes an infrared and visible image fusion framework based on generative models. This framework designs a multi-scale variational bottleneck encoder to extract structured low-level features and a physics-aware diffusion process to guide this extraction process. They provide extensive experiments to verify the effectiveness of this work across multiple fusion scenarios and in downstream semantic segmentation tasks.

**Questions:**

a)	To my understanding, MSRS serves as an enhanced version of the MFNet [IEEE IROS, 2017: 5108-5115.] dataset. Have the authors considered evaluating the proposed algorithm on MFNet to assess its performance on lower-quality data?
b)	In line 134, Clark's viewpoint is mentioned without a corresponding citation.
c)	Besides, please address the concerns in the Weakness part.

**Ethical Concerns:**

["NO or VERY MINOR ethics concerns only"]

**Final Justification:**

I am inclined to maintain my rating.

**Limitations:**

Yes.

**Paper Formatting Concerns:**

N/A.

**Quality:**

3

**Strengths And Weaknesses:**

Strengths:
1. The idea of revisiting generative infrared and visible image fusion from the perspective of human cognitive principles is novel.
2. The authors present theoretical constructs and theorems to substantiate the proposed method.
3. The authors conduct extensive evaluations, proving the advanced performance of this work.

Weaknesses：
1. In line 162, Why are only infrared images transformed? And what are the specific transformations applied? Providing detailed explanations would improve the clarity and readability of the manuscript.
2. In line 174, $\M_s$ represents mask weights that determine the importance of each feature. However, it is unclear how these weights are derived. Please provide a detailed explanation.
3. Also, the process of obtaining $\M_{heat}$ and $\M_{stru}$ should be described. Providing further detailed explanations would enhance the understanding of the overall method.
4. Have the authors evaluated the necessity of the aforementioned mask weights and analyzed their impact on the overall experimental results? It would be helpful if the authors could provide additional explanations to clarify this aspect.
5. In Table 5, the content in the last column appears to exceed the boundary. The authors may need to check the table formatting to ensure proper alignment.

---

> ### Author Rebuttal · Authors · 2025-07-30
>
> We sincerely thank you for the valuable comments and suggestions. We have carefully revised the manuscript according to your feedback. All modifications have been incorporated into the revised version. Below, we respond to each of your concerns in detail.
>
> **Response to Weakness 1:**
>
> In our framework, we deliberately apply transformation only to the infrared image while keeping the visible image unchanged. This design choice is primarily based on two considerations:
>
> (1) **Modality discrepancy**: Infrared images generally contain less structural detail and more semantic ambiguity compared to visible images due to inherent sensor limitations. Therefore, aligning the infrared modality towards the visible one — which typically exhibits more stable textures and clearer semantic content — helps to reduce cross-modal discrepancies and facilitates the learning of more consistent joint representations.
>
> (2) **Computational efficiency and robustness**: Aligning both modalities simultaneously would require computing and optimizing two separate transport plans, increasing training complexity and introducing potential inconsistencies between mappings. A one-sided transformation from infrared to visible ensures efficiency and stability during optimization.
>
> In practice, we compute a regularized optimal transport plan using the following formulation:
> $$
> \mathbf{P}^* = \arg\min_{\mathbf{P} \in \mathcal{U}(r, c)} \sum_{i,j} P_{ij} C_{ij} + \varepsilon \sum_{i,j} P_{ij} \log P_{ij},
> $$
> where $\mathcal{U}(r, c)$ denotes the set of doubly stochastic matrices with row and column marginals $r$ and $c$, $C_{ij}$ is the cost matrix computed by squared Euclidean distances between flattened pixel values of the infrared and visible images, and $\varepsilon$ is a regularization coefficient.
> After obtaining the transport plan $\mathbf{P}^*$, the infrared image $X \in \mathbb{R}^{B \times C \times H \times W}$ is first flattened to $X_{\text{flat}} \in \mathbb{R}^{B \times N \times C}$, where $N = H \times W$, and transformed by:
>
> $$
> X' = \mathbf{P}^* \cdot X_{\text{flat}},
> $$
>
> followed by reshaping back to $\mathbb{R}^{B \times C \times H \times W}$. The transformed infrared image $X'$ and the original visible image $Y$ are then fed into the variational bottleneck encoder for subsequent fusion modeling.
> We have added this explanation and implementation clarification in the revised manuscript for better clarity.
>
> **Response to Weakness 2:**
>
> Thank you for raising the questions regarding the derivation and effectiveness of the mask weights $M_s$ described in line 174. To clarify, the mask weights $M_s$ are computed as $M_s = \mathrm{sigmoid}(w_s)$, where $w_s \in \mathbb{R}^{1 \times C \times 1 \times 1}$ is a learnable parameter introduced at each encoder scale. This parameter is initialized from a standard Gaussian distribution using "torch.randn()" and registered as an “nn.Parameter". The sigmoid activation ensures that the mask values remain within a continuous range of $(0, 1)$, enabling soft, differentiable modulation of feature importance.
>
> **Response to Weakness 3:**
>
> We sincerely thank you for highlighting the importance of clarifying the construction process of the masks $M_{\text{heat}}$ and $M_{\text{stru}}$. These masks are explicit, non-learnable spatial priors designed based on statistically meaningful physical patterns inherent in the input modalities.
>
> *$M_{\text{heat}}$ is derived from the thermal intensity distribution of the infrared image. Specifically, we apply a $3 \times 3$ average pooling to obtain a local heat response map, then compare each pixel to the global mean value to construct a binary mask—pixels above the mean are set to 1, others to 0. This explicitly highlights thermally significant regions and models localized heat saliency.
>
> *$M_{\text{stru}}$ captures structural saliency in the visible image by applying a Laplacian operator to extract high-frequency edge responses (i.e., local second-order derivatives). This structural response map reflects intensity variations along object boundaries. We binarize it via thresholding at the mean, thereby identifying regions with prominent structural features.
>
> Both masks are dynamically updated at each reverse diffusion timestep, providing a time-varying guidance mechanism that adaptively emphasizes thermal or structural cues depending on the generation stage. This design enhances modality complementarity while reinforcing physical consistency and interpretability throughout the sampling trajectory.
>
> **Response to Weakness 4:**
>
> To address your concern, we conduct a detailed ablation study covering all three types of masks involved in our framework.
>
> Ablation 1 — W/O $M_s$:
>
> We remove the $M_s$ from the latent representation and observe clear performance degradation across multiple perceptual and structural metrics, confirming that $M_s$ is essential for filtering informative latent variables and reducing redundancy.
>
> Ablation 2 — W/O $M_{\text{heat}}$ & $M_{\text{stru}}$:
>
> We replace $M_{\text{heat}}$ and $M_{\text{stru}}$ with all-one masks, effectively removing their spatial selectivity. The performance drops consistently across key indicators related to detail and structure preservation (e.g., AG, SF, QSF), indicating that omitting spatial mask guidance weakens both visual quality and perceptual fidelity.
>
> The results are summarized in the revised ablation table (see Appendix), which is reproduced below for clarity. These findings demonstrate that each mask plays an indispensable role in its corresponding mechanism: $M_s$ in the bottleneck pathway, and $M_{\text{heat}}$, $M_{\text{stru}}$ in physically-guided image generation.
>
> The best results are highlighted in bold.
>
> || SD↑ | AG↑ | CC↑ | SCD↑ | EN↑ | SF↑ | Nabf↓ | DF↑ | QSF↑ |
> |---------|------|------|------|--------|------|-------|--------|------|-------|
> | W/O $M_s$ | 43.9840 | 7.1404 | 0.5026 | 1.7520| **7.1191** | 16.8872 | **0.0023** | 8.6568 | 0.4692 |
> | W/O $M_\text{heat}$ \& $M_\text{stru}$ | 39.1644 | 6.8207 | **0.5070** | 1.7492 | 6.9906 | 15.7408 | 0.0031 | 7.9742 | 0.3687 |
> | Ours | **47.7260** | **7.2112** | 0.4838 | **1.7673**| 7.0975 | **17.6254** | 0.0051 | **9.1041** | **0.5264** |
>
> **Response to Weakness 5:**
>
> We have revised the layout of Table 5 to ensure that the last column is properly aligned and fits within the page boundaries. Additionally, we have checked and adjusted the formatting of all other tables to ensure consistent alignment and presentation throughout the manuscript.
>
> **Response to Question a:**
>
> We sincerely thank you for raising this important point. In response, we have conducted additional experiments on the MFNet dataset. As shown in the results table below, our method retains strong robustness and generalization ability even under degraded sensor conditions.
>
> The best results are highlighted in bold, while the second-best results are marked with a † symbol.
>
> |Method|SD↑|AG↑|CC↑|SCD↑|EN↑|SF↑|Nabf↓|DF↑|QSF↑|
> |------|----|----|----|-----|----|----|-----|----|-----|
> |FusionGAN|23.8915|2.1156|0.5896†|1.2885|6.0584|6.4544|0.0068|2.5495|-0.4253|
> |NestFuse|39.9392|3.6089|0.5677|1.5369|6.6525†|11.4370|0.0024†|4.2067|-0.0540|
> |SwinFusion|40.0755|4.1712†|0.5538|1.5156|6.5820|12.8671†|0.0024†|4.8956†|0.0816†|
> |SegMiF|36.5946|3.0624|0.5601|1.4905|6.3983|9.1332|0.0062|3.4995|-0.2315|
> |SOSMaskFuse|43.7900|3.7247|0.5087|1.3222|6.6309|11.8681|0.0042|4.3045|-0.0053|
> |LRRNet|32.3548|2.9572|0.5356|1.3819|6.3787|9.1486|0.0066|3.4833|-0.2174|
> |STFNet|36.4268|3.0490|0.5554|1.4466|6.4178|9.0055|0.0033|3.3810|-0.2399|
> |CrossFuse|34.7190|3.2444|0.4836|1.1286|6.5573|9.9844|0.0081|3.8144|-0.1568|
> |Diff-IF|33.8229|3.5852|0.5467|1.3792|6.3762|11.4678|0.0025|4.2272|-0.0403|
> |CCF|28.3691|2.7630|**0.6077**|1.5403†|6.3183|8.9107|0.0041|3.4424|-0.2492|
> |MMAE|**53.9142**|3.3900|0.3530|0.8330|6.5531|11.5492|0.0042|3.8339|0.0505|
> |LFDT-Fusion|35.2642|3.6937|0.5434|1.3966|6.4728|11.4020|0.0033|4.2848|-0.0428|
> |Ours|46.3864†|**6.0428**|0.5454|**1.5466**|**6.9013**|**17.4240**|**0.0007**|**7.2109**|**0.4782**|
>
>
> **Response to Question b:**
>
> We sincerely apologize for the omission. We have added the citation to Clark's work in the revised manuscript (Line 134):
>
> Clark, A. "Whatever next? Predictive brains, situated agents, and the future of cognitive science." Behavioral and Brain Sciences, 2013, 36(3): 181–204.
>
>
> **Response to Question c:**
>
> We appreciate your advice and have carefully addressed the concerns raised in the Weaknesses section.

---

> > ### Comment · Reviewer_cS6j · 2025-08-04
> >
> > Thank you for your efforts during the rebuttal. However, I still have concerns about the explanation of how  $\M_{heat}$ and $\M_{stru}$  are obtained. The way you generate binary masks based on the thermal intensity distribution of infrared images and the gradient distribution of infrared images seems rather simplistic. For instance, not all high-saliency radiation represents valid information (such as roads heated by sunlight), and not all high gradients indicate valid information, like those caused by noise.

---

> > > ### Author Response · Authors · 2025-08-04
> > >
> > > Thank you for your further comments. We fully understand your concern that masks generated based on simple rules may not accurately capture real and meaningful regions.  In our method, the purpose of $M_{\text{heat}}$ and $M_{\text{stru}}$ is not to precisely select target regions, but to introduce modality-specific physical guidance cues that assist the generation process in evolving toward structural consistency and physical plausibility. To enhance the distribution learning capability of the diffusion model and reduce its dependence on data, we incorporate a Time-varying Physical Guidance (TPG) mechanism during sampling. Consequently, the risk of region misjudgment caused by static masks  is gradually mitigated through a dynamic attenuation strategy.
> > >
> > > We also acknowledge that more complex or learnable masking mechanisms may further enhance robustness. However, our current design strikes a balance among interpretability, physical consistency, and controllability, making it more suitable for unsupervised image fusion tasks.

---

### Official Review · Reviewer_bhfu · 2025-06-30

**Clarity:** 3
**Significance:** 3
**Originality:** 4
**Rating:** 5
**Confidence:** 5

**Summary:**

This paper proposes a generative infrared and visible image fusion method based on human cognitive laws, named HCLFuse. By integrating a variational bottleneck encoder and a physics-guided diffusion model, it enhances the structural consistency and detail quality of fused images.

**Questions:**

Here are several problems and suggestion for authors to improve their paper.

(1) In Eq.6, how are the mask weights $M_s$ generated? It is recommended to provide a sufficient explanation.

(2) The Method section lacks an overview of the entire HCLFuse process. It is recommended that the authors add a textual description or pseudocode in the main text or appendix to outline the complete workflow from inputs (X, Y) to output Z.

(3) VIF (Visual Information Fidelity) demonstrates greater advantages in perceptual quality assessment and is more widely applied. It is recommended to replace the QSF metric with VIF, which is also more relevant to the "Human Cognitive Laws" mentioned in the paper's title.

(4) The ablation experiments are insufficiently comprehensive, as the authors do not validate the role of the heat conduction constraint. It is recommended to add relevant experiments to address this gap.

(5) In the ablation experiments subsection, what is DDIM? What is "deterministic sampling"? It is recommended to provide clear explanations.

**Ethical Concerns:**

["NO or VERY MINOR ethics concerns only"]

**Final Justification:**

The authors have addressed my concerns and revised the paper as suggested, improving its quality. It is acceptable.

**Limitations:**

yes

**Paper Formatting Concerns:**

No formatting issues.

**Quality:**

3

**Strengths And Weaknesses:**

This paper innovatively introduces human cognitive laws into generative image fusion, proposing a novel HCLFuse framework. Its unique innovations lie in the design of a multi-scale variational bottleneck encoder and a time-varying physics-guided mechanism, which distinguish it from both traditional methods and existing generative approaches.

HCLFuse demonstrates high fusion quality: it achieves optimal performance across multiple quantitative metrics (e.g., AG, SF, DF, QSF) on datasets such as MSRS, TNO, and FMB. In qualitative evaluations, it effectively preserves complementary features, exhibits clear details, and ensures strong structural consistency, outperforming various comparative methods.

However, the theoretical framework is overly abstruse, imposing certain requirements on readers' prior knowledge. This may hinder the accessibility of the research findings to a broader audience.

---

> ### Author Rebuttal · Authors · 2025-07-30
>
> We sincerely thank you for your constructive feedback and recognition of the contributions of our work. Below, we address each of the reviewer’s concerns in detail.
>
> **Response to Weakness:**
>
> To address the concern that parts of the theoretical presentation may hinder accessibility for a broader audience,
>
> (1) We have carefully revised the manuscript based on your comments, improving unclear expressions and enhancing clarity throughout;
>
> (2) We have added more references to support the foundational theories involved;
>
> (3) We will release the complete code in due course to promote transparency and reproducibility.
>
> **Response to Question 1:**
>
> Thank you for the comment regarding the mask weights $M_s$ in Eq.~(6), which represent a critical component of the proposed design. At each encoder scale, a structurally defined learnable parameter $w_s \in \mathbb{R}^{1 \times C \times 1 \times 1}$, where $C$ denotes the number of feature channels, is introduced. The parameter is initialized from a standard Gaussian distribution using “torch.randn()" and registered as an "nn.Parameter" in the model. During forward propagation, $M_s = \mathrm{sigmoid}(w_s)$, forming a differentiable soft mask applied to the feature map. Throughout training, $w_s$ is optimized under the guidance of the information bottleneck loss $\mathcal{L}_{\text{VBE}}$, functioning as an auxiliary modulation mechanism that enhances the selective suppression of redundant information within the latent representation.
>
> To address your concern, we conduct a dedicated ablation study to examine the individual and joint contributions of the mask mechanism and the information bottleneck loss. The results clearly show that removing either component degrades performance, while the best results are consistently achieved only when both are jointly applied. This highlights the necessity of their integration in realizing effective latent compression.
>
> **Response to Question 2:**
>
> Thank you for this helpful suggestion. We have added an overview of the entire HCLFuse process at the beginning of the Method section. In addition, we have included a pseudocode description of the complete workflow in the appendix to further enhance reproducibility and clarity. To facilitate your review, we present below the textual description as it appears in the Method section:
>
> "In HCLFuse, let the infrared image domain be denoted as $X$ and the visible image domain as $Y$, with joint observations satisfying $(x, y) \sim p_{x,y}$. We construct a fusion mapping $F(X, Y) \rightarrow Z$ to generate the fused latent representation $z = F(x, y) \in Z$, where $Z \subseteq \mathbb{R}^d$ denotes the fusion space. Ideally, the fused representation should effectively preserve the complementary information from both modalities while suppressing redundant noise. To achieve this, HCLFuse first employs a multi-scale, mask-regulated variational bottleneck encoder to compress and model the latent representation $z$. Before encoding, an optimal transport-based mapping is applied to improve the optimization lower bound, encouraging $z$ to contain modality-discriminative content in the unsupervised setting. Subsequently, $z$ is refined through a reverse-time diffusion generation process, where physically guided constraints are dynamically injected at each denoising timestep to guide the evolution of latent features. Finally, the optimized latent representation is decoded to produce the fused image."
>
> **Response to Question 3:**
>
> Regarding your suggestion on the use of VIF, we agree that VIF is a widely adopted full-reference metric for perceptual quality assessment and aligns well with the "Human Cognitive Laws" emphasized in our title. In response, we have included additional experiments and reported VIF results for completeness. However, VIF is a full-reference metric that relies on source images, which conflicts with the goal of generative fusion. Our method is designed to synthesize richer and more complete information by integrating both modalities, and as illustrated in Figures 3, 4, and 6–9 in the paper, the fused outputs often surpass the original inputs in clarity, detail, and semantic structure. As a result, using source images as references tends to underestimate the perceptual quality of the generated fusion results.
>
> To address this, we have further introduced two no-reference perceptual metrics—PIQE [IEEE NCC, 2015: 1–6.] and BRISQUE [IEEE TIP, 2012: 4695–4708.]—which are well-suited for evaluating visual quality without requiring reference images. They capture distortions and unnatural statistics in a way that correlates with human perception. As shown in the table below, although our method achieves a moderate score on the VIF metric, it demonstrates superior performance on both the PIQE and BRISQUE metrics, highlighting its perceptual quality in image generation.
>
> The best results are highlighted in bold, while the second-best results are marked with a † symbol.
>
> |Method|MSRS|dataset||FMB|dataset||TNO|dataset||MFNet|dataset||
> |--------------|------------------|-------|--------|------------------|-------|--------|------------------|-------|--------|------------------|-------|--------|
> ||VIF↑|PIQE↓|BRISQUE↓|VIF↑|PIQE↓|BRISQUE↓|VIF↑|PIQE↓|BRISQUE↓|VIF↑|PIQE↓|BRISQUE↓|
> |FusionGAN|0.4627|44.12|35.54|0.4452|38.19|28.46|0.4220|23.09|27.80|0.5563|38.69|33.24|
> |NestFuse|0.9099|41.86|37.51|0.8818|33.86|21.75|0.8651|22.78|24.69|**1.0281**|38.60|33.10|
> |SwinFusion|0.9119|38.14|37.55|0.8965†|30.36|24.85|0.7503|20.66|24.11†|0.9410|31.85|32.66|
> |SegMiF|0.6239|42.08|34.44|0.6722|37.48|22.02|0.7028|23.35|25.41|0.6699|44.90|34.38|
> |SOSMaskFuse|0.8585|44.77|40.06|**0.9628**|32.53|22.60|0.8765†|21.64|25.89|0.9405|40.79|35.12|
> |LRRNet|0.5680|29.96|30.82|0.6324|33.79|**19.41**|0.5612|**16.42**|29.52|0.7167|30.94†|31.17|
> |STFNet|0.8463|54.53|43.03|0.6476|51.92|30.63|0.7205|35.98|37.47|0.8524|51.42|38.72|
> |CrossFuse|0.8374|33.08|33.59|0.8142|30.35|23.82|0.7365|20.30†|26.88|0.8669|34.46|33.70|
> |Diff-IF|**1.0417**|32.73|31.51|0.8717|29.14†|20.14†|0.8433|20.63|**22.83**|0.9191|36.26|33.04|
> |CCF|0.6847|**17.34**|26.38†|0.5252|49.22|34.47|0.5503|31.04|36.63|0.7539|20.22|29.50†|
> |MMAE|0.8090|33.18|31.25|0.8526|33.39|22.28|0.8229|25.22|26.16|0.8061|44.60|39.09|
> |LFDT-Fusion|1.0296†|38.86|32.37|0.7291|33.12|22.71|**0.8774**|23.42|25.18|0.9713†|40.76|34.15|
> |**Ours**|0.8540|25.19†|**26.22**|0.7629|**26.42**|33.84|0.6137|25.71|29.26|0.9261|**24.13**|**27.99**|
>
> **Response to Question 4:**
>
> We sincerely thank you for pointing out the lack of explicit validation for the heat conduction constraint. To address this concern, we have conducted a comprehensive ablation study targeting each physical constraint. The results are summarized in the updated Table:
>
> The best results are highlighted in bold.
> ||$\Phi_\text{heat}$|$\Phi_\text{stru}$|$\Phi_\text{con}$|SD↑|AG↑|CC↑|SCD↑|EN↑|SF↑|Nabf↓|DF↑|QSF↑|VIF↑|PIQE↓|BRISQUE↓|
> |----------------------------|------|------|------|---------|---------|---------|---------|---------|---------|---------|---------|---------|---------|---------|-----------|
> |A|✗|✗|✗|43.1107|6.6147|0.5028|1.8086|6.9285|15.9481|0.0040|8.1256|0.3838|0.5196|20.3652|37.2477|
> | B | ✓    | ✗    | ✗    | 42.0447 | 6.8718  | 0.5035  | 1.8076  | 7.0436  | 16.5597 | **0.0029**  | 8.5954  | 0.4480  | 0.5164  | **18.3564** | 35.0540   |
> | C | ✓    | ✓    | ✗    | 41.2477 | 6.7049  | **0.5080**  | **1.8183**  | 7.0248  | 16.5085 | 0.0034  | 8.6943 | 0.4319  | 0.5054  | 21.2138 | 35.3540   |
> | Ours | ✓    | ✓    | ✓    | **47.7260** | **7.2112**  | 0.4838  | 1.7673  | **7.0975**  | **17.6254** | 0.0051  | **9.1041**  | **0.5264**  | **0.6137**  | 25.7097 | **29.2594** |
>
> Experimental Analysis:
>
> From A → B, introducing only the heat conduction constraint $\Phi_{\text{heat}}$ already improves several perceptual indicators (e.g., AG, EN, SF), and achieves the lowest Nabf and PIQE.
>
> From B → C, adding the structure preservation constraint $\Phi_{\text{stru}}$ further enhances structural similarity (e.g.,SCD), showing that $\Phi_{\text{stru}}$ helps maintain sharpness and edge consistency.
>
> From C → Ours, the inclusion of the physical consistency constraint $\Phi_{\text{con}}$ results in the best overall performance across almost all metrics, with notable gains in SD, AG, EN, SF, and VIF. This underlines the complementary and synergistic role of the three constraints.
>
> These results collectively validate the effectiveness of each individual physical constraint, and further demonstrate that only the full combination of all three can maximize overall fusion performance.
>
> **Response to Question 5:**
>
> We sincerely thank you for pointing out the lack of clarity regarding DDIM. This is indeed an important component that was not sufficiently explained in the original version. DDIM (Denoising Diffusion Implicit Models) [Song et al., ICLR 2021] is a deterministic variant of the standard diffusion model DDPM(Denoising Diffusion Probabilistic Models) [NeurIPS, 2020: 6841–6853]. In our framework, DDIM serves as the core generative module. In the ablation setting "W/O DDIM", we remove the entire diffusion process and obtain the fused image directly through an encoder-decoder structure. As shown in Table 2 and Figure 5 in the paper, this leads to a loss of fine details and noticeable drops in several metrics, confirming that the generative process provides essential complementary information for high-quality fusion. Additional explanations and the relevant citation have been added in the revised manuscript for clarity.

---

> > ### Comment · Reviewer_bhfu · 2025-08-06
> >
> > The authors' responses have addressed my concerns, and I hold the view that this paper deserves acceptance.

---

> > > ### Author Response · Authors · 2025-08-06
> > >
> > > Thank you very much for your thoughtful feedback and for recognizing the improvements in our revised responses. We sincerely appreciate your support throughout the review process.

---

### Official Review · Reviewer_pn8n · 2025-07-05

**Clarity:** 3
**Significance:** 2
**Originality:** 3
**Rating:** 4
**Confidence:** 3

**Summary:**

This paper proposes a novel infrared and visible image fusion method, HCLFuse, aiming to address the limitations of existing generative fusion techniques by incorporating "human cognitive laws". The paper designs a method that integrates a variational bottleneck encoder and a physics-guided diffusion model. It demonstrates excellent quantitative and qualitative performance on multiple benchmark datasets, with notable improvements in downstream semantic segmentation tasks. The writing is clear, the experimental section is well-organized, and the ablation studies are fairly comprehensive.

**Questions:**

Please refer to the Weaknesses.

**Ethical Concerns:**

["NO or VERY MINOR ethics concerns only"]

**Final Justification:**

The author addressed all of my concerns satisfactorily. I suggest accepting this paper.

**Limitations:**

Yes

**Paper Formatting Concerns:**

Null

**Quality:**

3

**Strengths And Weaknesses:**

Strengths:

1. The paper clearly outlines the challenges of image fusion, the limitations of existing methods, and its proposed solutions. The presented framework structure is also intuitive and easy to follow.

2. The paper presents SOTA performance on several infrared and visible image fusion datasets (MSRS, TNO, FMB). Particularly in qualitative results, HCLFuse demonstrates impressive capabilities in detail preservation, structural consistency, and cross-modal information integration.

Weaknesses:

1. The paper elevates mathematical tools like IB and OT to the status of "human cognitive laws". Applying them to image fusion, even if their functionality "resembles" certain aspects of human cognition (e.g., information filtering, modality alignment), does not directly equate to them being actual manifestations of "human cognitive laws" .

2. The "physical law" constraints claimed in the paper (Heat Conduction, Structure Preservation, Physical Consistency), in the technical implementation, appear to be common image processing priors or regularization terms. For instance, using the Laplacian operator to promote smoothness is essentially a high-pass filtering or edge detection operation, commonly used in image denoising or enhancement.

3. Despite the excellent model performance, the core components (VBE as a variant of VAE, introducing prior knowledge into diffusion models) are incremental contributions.

---

> ### Author Rebuttal · Authors · 2025-07-30
>
> We sincerely thank you for your insightful and constructive comments regarding the core contributions and scientific novelty of our work. Your thoughtful observations prompted us to more clearly articulate the essence of our contribution. We would like to clarify that the central innovation of this work does not lie in simply applying existing mathematical tools, but in redefining unsupervised image fusion under limited-data settings as a cognition-inspired generative problem. To this end, we present a coherent modeling strategy spanning theoretical motivation, architectural design, and mathematically grounded implementation. This reflects our shift from traditional heuristic fusion pipelines to an interpretable and principled generative approach guided by cognitive processes. Specifically, our innovations span three levels:
>
> *A cognition-inspired image fusion framework is proposed by reformulating the fusion task as a generative problem, aiming to obtain high-quality fused images beyond the limitations of individual modalities.
>
> *An unsupervised information mapping and quantization theory is proposed to characterize the trade-off between multi-source information compression and discriminability, based on which a multi-scale variational bottleneck encoder is designed to extract task-relevant low-level feature representations.
>
> *A dynamically evolving physics-guided mechanism is incorporated into the diffusion process, where temporally progressive physical priors enhance structural perception and generalization, effectively reducing dependence on large-scale training data.
>
> To address your concerns in detail, we respond to your comments point by point below.
>
> **Response to Weakness 1:**
>
> Regarding your concern about the use of the phrase “human cognitive laws” , we fully acknowledge that this wording may have unintentionally overstated our intention. We have revised the manuscript to clarify that our framework is guided by **cognition-inspired modeling principles**, rather than literal human cognitive laws. Specifically, we are inspired by how humans process multi-source information: filtering redundant content, attending to salient cues, and forming abstract internal representations. The Information Bottleneck (IB) principle models the selective compression process inherent in such cognitive abstraction, while Optimal Transport (OT) provides a tractable and unsupervised alternative objective that enables alignment and regularization of cross-modal information. This theoretical foundation is detailed in Theorem 1. Together, these tools are not themselves “cognitive laws,” but are used to instantiate a formal, interpretable, and optimizable framework that reflects cognition-like information processing. We hope this can clarifies our conceptual positioning.
>
> **Response to Weakness 2:**
>
> Thank you for raising this important point. We fully agree that individual components such as Laplacian smoothing have been used as traditional image priors. However, the novelty of our work does not lie in each constraint’s isolated form, but in how they are dynamically and sequentially embedded into the generative diffusion process through a mechanism we term Time-varying Physical Guidance (TPG).
> This mechanism introduces three physically-inspired constraints—heat conduction ($\Phi_{\text{heat}}$), structure preservation ($\Phi_{\text{stru}}$), and cross-modal consistency ($\Phi_{\text{con}}$)—that are:
>
> Applied in sequence to the predicted sample at each reverse diffusion timestep;
>
> Modulated by a decaying coefficient $\lambda(t)$ that mimics coarse-to-fine reasoning;
>
> Designed to stabilize generation and guide physical structure, not merely smooth pixels.
>
> This guidance process mimics the cognitive behavior of humans resolving ambiguous information by applying strong structural priors early and gradually relaxing them as certainty increases.
>
> **Response to Weakness 3:**
>
> While our Variational Bottleneck Encoder (VBE) adopts a variational formulation, it is not a simple extension of the conventional VAE. The key distinction lies in both its theoretical foundation and functional role. VBE is directly motivated by our theoretical result in Theorem 1, where we reformulate unsupervised image fusion as an information quantization problem, leading to a principled, task-aware latent abstraction objective that fundamentally differs from the reconstruction-driven design of standard VAEs.
>
> More importantly, VBE serves as a functional bridge between mutual information optimization and the generative process. It provides the diffusion model with a structurally coherent and task-relevant latent space, effectively relieving the burden of handling complex cross-modal alignment and selection. This allows the diffusion model to focus on modeling high-quality fused outputs guided by the compressed priors. Notably, most existing diffusion-based methods focus solely on improving the architecture itself, while overlooking the task relevance and informational clarity of the latent inputs. Our approach addresses this upstream bottleneck optimizing the content of the latent space, thereby enhancing both the generative ability and fusion quality.
>
> **Supplementary Experiments:**
>
> To further support your concerns on originality and effectiveness, we have conducted comprehensive additional experiments:
>
> (1) An ablation analysis of both the internal structure and overall design of VBE demonstrates its critical role in compact encoding;
>
> (2)A fine-grained ablation study on each physical constraint confirms their individual benefits and complementary synergy;
>
> (3)A hyperparameter study over $\beta$, $\lambda_{\text{heat/stru/con}}$, and decay factor $\gamma$ validates the robustness of our design choices;
>
> (4) Comparative evaluations have been expanded by incorporating additional top-tier conference methods, resulting in a total of seventeen compared approaches (e.g., TarDAL [CVPR 2022], DDFM [ICCV 2023], Text-DiFuse [NeurIPS 2024], Text-IF [CVPR 2024], GIFNet [CVPR 2025]), and are conducted across twelve quality metrics, including spatial detail, structural preservation, and perceptual quality;
>
> (5) The MFNet dataset has been additionally introduced, and the experimental results further verify the generalization ability of the proposed method under low-quality scenarios.
>
> Thank you once again for your thoughtful comments, which have significantly improved our presentation and helped sharpen the contribution of our work. We hope the revised version addresses your concerns and clarifies the novelty and value of our approach.

---

> > ### Comment · Reviewer_pn8n · 2025-08-06
> >
> > Sorry for this late response. The author addressed all of my concerns. I will raise my score.

---

> > > ### Author Response · Authors · 2025-08-07
> > >
> > > We sincerely thank you for your kind follow-up and for taking the time to re-evaluate our work. Your thoughtful engagement and willingness to update the score are deeply appreciated.

---

### Decision · Program_Chairs · 2025-09-17

**Decision:**

Accept (spotlight)

**Comment:**

This paper introduces a novel generative framework for infrared–visible image fusion inspired by principles of human cognition. Reviewers responded very positively, highlighting its strengths and requesting minor clarifications that were effectively addressed in the rebuttal. Consequently, all reviewers recommended acceptance.